# Defining the location of promoter-associated R-loops at near-nucleotide resolution using bisDRIP-seq

Jason G Dumelie, Samie R Jaffrey*

Department of Pharmacology, Weill Cornell Medical College, Cornell University, New York, United States

**Abstract** R-loops are features of chromatin consisting of a strand of DNA hybridized to RNA, as well as the expelled complementary DNA strand. R-loops are enriched at promoters where they have recently been shown to have important roles in modifying gene expression. However, the location of promoter-associated R-loops and the genomic domains they perturb to modify gene expression remain unclear. To resolve this issue, we developed a bisulfite-based approach, bisDRIP-seq, to map R-loops across the genome at near-nucleotide resolution in MCF-7 cells. We found the location of promoter-associated R-loops is dependent on the presence of introns. In intron-containing genes, R-loops are bounded between the transcription start site and the first exon-intron junction. In intronless genes, the 3' boundary displays gene-specific heterogeneity. Moreover, intronless genes are often associated with promoter-associated R-loop formation. Together, these studies provide a high-resolution map of R-loops and identify gene structure as a critical determinant of R-loop formation.

DOI: https://doi.org/10.7554/eLife.28306.001

*For correspondence:
srj2003@med.cornell.edu

**Competing interests:** The authors declare that no competing interests exist.

## Introduction

R-loops are nucleic acid structures in which a strand of RNA is hybridized to a strand of DNA, while the other strand of DNA is looped out. Recent techniques for genome-wide mapping of R-loops revealed that promoter regions are enriched in R-loops (*Ginno et al., 2012*). The presence of R-loops in promoter regions raises the possibility that they may regulate gene expression. Indeed, more recent studies provided evidence that R-loops in these critical regions can alter histone modifications and are associated with changes in gene transcription (*Chen et al., 2015*; *Colak et al., 2014*; *Sun et al., 2013*).

A major unanswered question is the precise location of these R-loops. The location of an R-loop in a gene is likely to dictate how that R-loop can impact promoter function. This is because eukaryotic promoter regions contain multiple functional domains that have distinct roles in transcription, including transcription start sites, transcription factor-binding sites, exon-intron junctions, CpG islands, and nucleosome-associated DNA (*Lenhard et al., 2012*). The location of an R-loop within the promoter could influence transcription by disrupting or enhancing protein recruitment to any of these sites. Thus, understanding the precise location of R-loops can provide insight into how R-loops affect gene transcription.

A major barrier to discovering the exact location of R-loops is the low resolution of current genome-wide R-loop mapping methods like DRIP-seq (DNA-RNA immunoprecipitation sequencing) (*Ginno et al., 2012*). This method uses the S9.6 antibody which binds RNA-DNA hybrids (*Boguslawski et al., 1986*). With this antibody, genome-wide R-loop maps are created by immunoprecipitating and sequencing the genomic fragments containing RNA-DNA hybrids (*Ginno et al., 2012*; *Sanz et al., 2016*; *Stork et al., 2016*). However, DRIP-seq does not discriminate between the

**eLife digest** Genes contain coded instructions for making proteins. When the cell needs to use a gene, molecular machinery assembles near the start of the gene in regions called promoters. Part of this machinery then reads along the gene, making a copy of the code in the form of a DNA-like molecule called RNA. These RNAs typically contain regions called exons, which carry the instructions, interspersed with spacer regions called introns. As RNAs are made they are 'spliced' to chop out the introns, leaving behind the final instructions.

Most DNA exists in a double helix shape with two connected DNA strands, but the regions near the start of genes often contain structures called R-loops. In these structures, one strand of the DNA partners up with a single strand of RNA, forcing the other strand to bulge out on its own. Their location at gene promoters indicates that R-loops could change the cell's use of genes by impacting the machines that assemble near the start of genes. However, R-loops are not well understood. A major barrier to understanding the role of R-loops is that we do not know exactly where they are with respect to the start of genes.

Dumelie and Jaffrey now report a new method to map R-loops almost to the resolution of single letters of the DNA code – a method which they called bisDRIP-seq. The approach extends an existing technique called DRIP-seq, which uses antibodies to capture DNA sequences stuck to strands of RNA. It can find R-loops, but it cannot tell the difference between the loop itself and the DNA surrounding it. The new technique uses a chemical called bisulfite to alter the DNA letters. It only affects the loop of the R-loop because the RNA shields the other strand. Sequencing then pinpoints the modified letters, revealing the exact location of the loop.

For human cells grown in the laboratory, the technique found that R-loops form between the start of the gene and its first intron. Some genes do not have any introns, and in these cases, the R-loops extended deep into the code. Most human genes have only a small amount of DNA between the start site and the first intron, which may act to limit the effect of R-loops in these genes.

This new technique allows the high-resolution study of R-loops, and could help to reveal their role in regulating genes. Abnormal R-loops have already been linked to a small set of human diseases like fragile-X syndrome. As the tools to study R-loops improve, it is possible that scientists will make connections to other diseases. In time, improved understanding of these structures could lead to better diagnosis, and eventually treatment, for these conditions.

DOI: https://doi.org/10.7554/eLife.28306.002

R-loop sequence and the surrounding non-R-loop sequence. Therefore, the exact boundaries of R-loops cannot be resolved using DRIP-seq.

Based on DRIP-seq and similar R-loop mapping methods, promoter-associated R-loops are thought to form within a few kilobases downstream of the transcription start site (*Chédin, 2016*). From these low-resolution experiments, it is not clear if R-loops have specific boundaries or if they are relatively amorphous structures that lack well-defined boundaries.

To understand where R-loops are positioned in genomic promoter regions, we developed bis-DRIP-seq (bisulfite-DNA-RNA immunoprecipitation sequencing). bisDRIP-seq is an approach to map R-loops at near-nucleotide resolution throughout the genome. In this approach, we use bisulfite to selectively convert cytosine residues into uracil residues within genomic DNA regions that contain single-stranded DNA. We then identify single-stranded regions likely to be in R-loops based on pref- erential labeling of one strand of DNA and the requirement that the labeling be transcription depen- dent. Remarkably, we find that promoter-associated R-loops are typically bounded by the transcription start site and the first exon-intron junction in intron-containing genes. Thus, we find that the maximum size of promoter-associated R-loops is controlled by the location of the first exon- intron junction in intron-containing genes. We also identify prominent promoter-associated R-loop forming regions in intronless genes, including *MALAT1*, *NEAT1*, and the replication-dependent his- tone genes. In some of these genes, the R-loops are associated with well-defined 3' boundaries that are located within the gene body. Thus our high-resolution map of promoter-associated R-loops

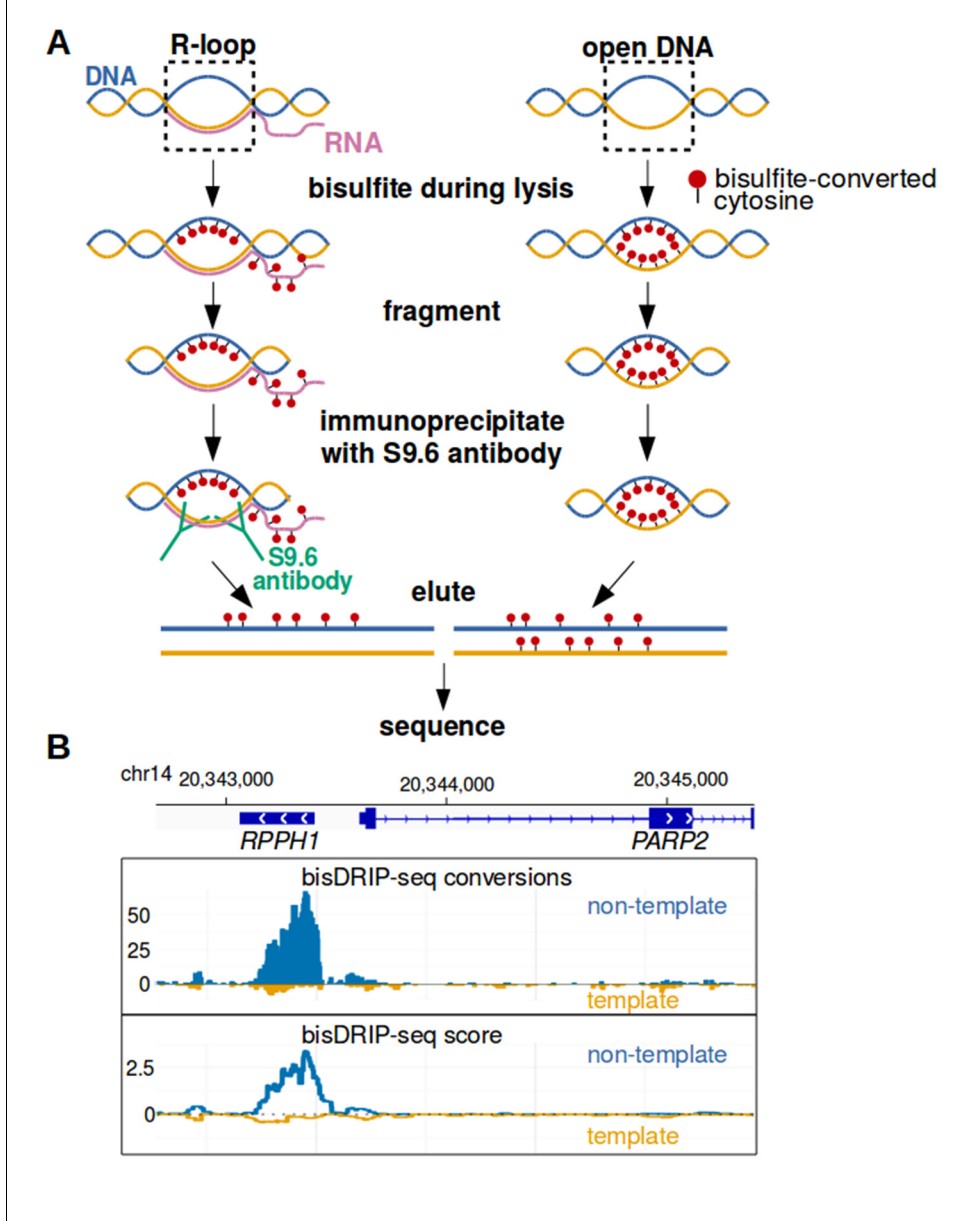

**Figure 1.** Near-nucleotide resolution mapping of R-loops. (**A**) Diagram illustrating the work-flow of the bisDRIP-seq protocol. For DNA molecules, one DNA strand is shown in blue, while the other strand is shown in orange. Red dots indicate the location of bisulfite-induced cytosine-to-uracil conversions. (**B**) A high number of cytosine conversions and high bisDRIP-seq scores are observed at a specific gene locus. bisDRIP-seq cytosine conversions and high bisDRIP-seq scores are specifically observed on one strand of the *RPPH1* gene. In these plots, the template strand refers to the strand used for *RPPH1* transcription, rather than the template strand used for *PAPR2* transcription. In the top plot, cytosine-to-uracil conversions were mapped to the genomic region surrounding *RPPH1*. The number of conversions on the template strand and non-template strand were plotted below the x-axis (orange) or above the x-axis (blue), respectively. Shown are the total number of conversions observed in all bisDRIP-seq samples (n = 13). In the lower plot, the bisDRIP-seq scores were mapped to the genomic region surrounding *RPPH1* (mean bisDRIP-seq score from n = 13 samples). The bisDRIP-seq score on the template strand and non-template strand were plotted below the x-axis (orange) or above the x-axis (blue), respectively. As can be seen, the genomic region containing *RPPH1* had both a high number of cytosine-to-uracil conversions and high bisDRIP-seq scores specifically on the non-template strand. By contrast, the *PARP2* gene had minimal bisDRIP-seq conversions and low bisDRIP-seq scores. Source code for bisDRIP-seq score calculation can be found in *Source code 1*.

DOI: https://doi.org/10.7554/eLife.28306.003

*Figure 1 continued on next page*

*Figure 1 continued*

The following figure supplements are available for figure 1:

**Figure supplement 1.** bisDRIP-seq identifies regions of single-stranded DNA.
DOI: https://doi.org/10.7554/eLife.28306.004
**Figure supplement 2.** bisDRIP-seq scores are sensitive to RNase H.
DOI: https://doi.org/10.7554/eLife.28306.005
**Figure supplement 3.** bisDRIP-seq scores of promoter regions were correlated between samples.
DOI: https://doi.org/10.7554/eLife.28306.006

---

defines the boundaries of R-loops and suggests a role for first exon length in regulating the formation of R-loops.

## Results

### bisDRIP-seq concept

It is not yet possible to define the exact size and location of R-loop-forming regions on a genome-wide scale. In contrast, it is possible to determine the exact location of a specific R-loop-forming region in a specific gene using a previously developed bisulfite mapping approach (*Yu et al., 2003*). Essentially, this approach involved treating genomic DNA with bisulfite under non-denaturing conditions. Bisulfite specifically causes cytosine-to-uracil conversions in the single-stranded DNA portion of R-loops. On the other hand, cytosines in the RNA-DNA hybrid portion of the R-loop are protected from bisulfite conversion. Sequencing of both strands then revealed the location and strand in which cytosines were converted to uracils. The presence of an R-loop was identified by showing that the converted cytosines occurred primarily on one of the two strands of DNA (*Yu et al., 2003*). The location of bisulfite-induced conversions on a single strand of DNA was then used to define the boundaries of the R-loop at near-nucleotide resolution.

This use of bisulfite to detect single-stranded regions of DNA contrasts with the use of bisulfite in mapping 5-methylcytosine in DNA. For 5-methylcytosine mapping, DNA is denatured into single strands and then all cytosines are converted to uracils, while 5-methylcytosine is poorly converted (*Frommer et al., 1992*). Thus, while mapping 5-methylcytosine involves searching for unconverted cytosines, mapping single-stranded DNA involves detection of converted cytosines.

The use of bisulfite to map R-loops on a genome-wide scale poses several significant challenges. First, non-physiological R-loop formation, removal, expansion, or contraction can occur between the time of lysis and the time when bisulfite is used to mark single-stranded DNA (*Kaback et al., 1979*; *Landgraf et al., 1996*). Second, high sequencing depth is needed to search for converted cytosines across the entire genome (*Sims et al., 2014*). Finally, bisulfite causes infrequent, but measurable, background conversions in double-stranded DNA (*Yu et al., 2003*), which can generate a significant amount of noise on a genome-wide scale.

To map R-loops at near-nucleotide resolution, we developed a genome-wide bisulfite-based approach called bisDRIP-seq (*Figure 1A*). This method incorporates steps to overcome each of the challenges listed above. In this approach, cells are lysed in the presence of bisulfite and SDS. By including bisulfite during lysis, genomic DNA structures have as little time as possible to change conformation prior to the bisulfite modification of cytosines in single-stranded DNA regions. To overcome the problem of needing high sequence coverage, the S9.6 antibody (*Boguslawski et al., 1986*) is used to enrich for R-loops. The S9.6 antibody has high affinity for RNA-DNA hybrids and lower affinity for other double-stranded RNA sequences (*Phillips et al., 2013*). Thus, after genomic DNA is sheared using restriction digestion, the S9.6 antibody enriches the bisulfite-modified R-loops for sequencing analysis.

Finally, to overcome the problem of stochastic cytosine conversions, a computational pipeline was developed to identify single-stranded regions using bisDRIP-seq data. This pipeline was developed to identify regions with high concentrations of cytosine-to-uracil conversions. This computational pipeline also identifies single-stranded regions that are likely to contain R-loops as opposed to other single-stranded DNA structures in the genome. Additionally, the pipeline was designed to reveal the specific strand orientation of R-loops.

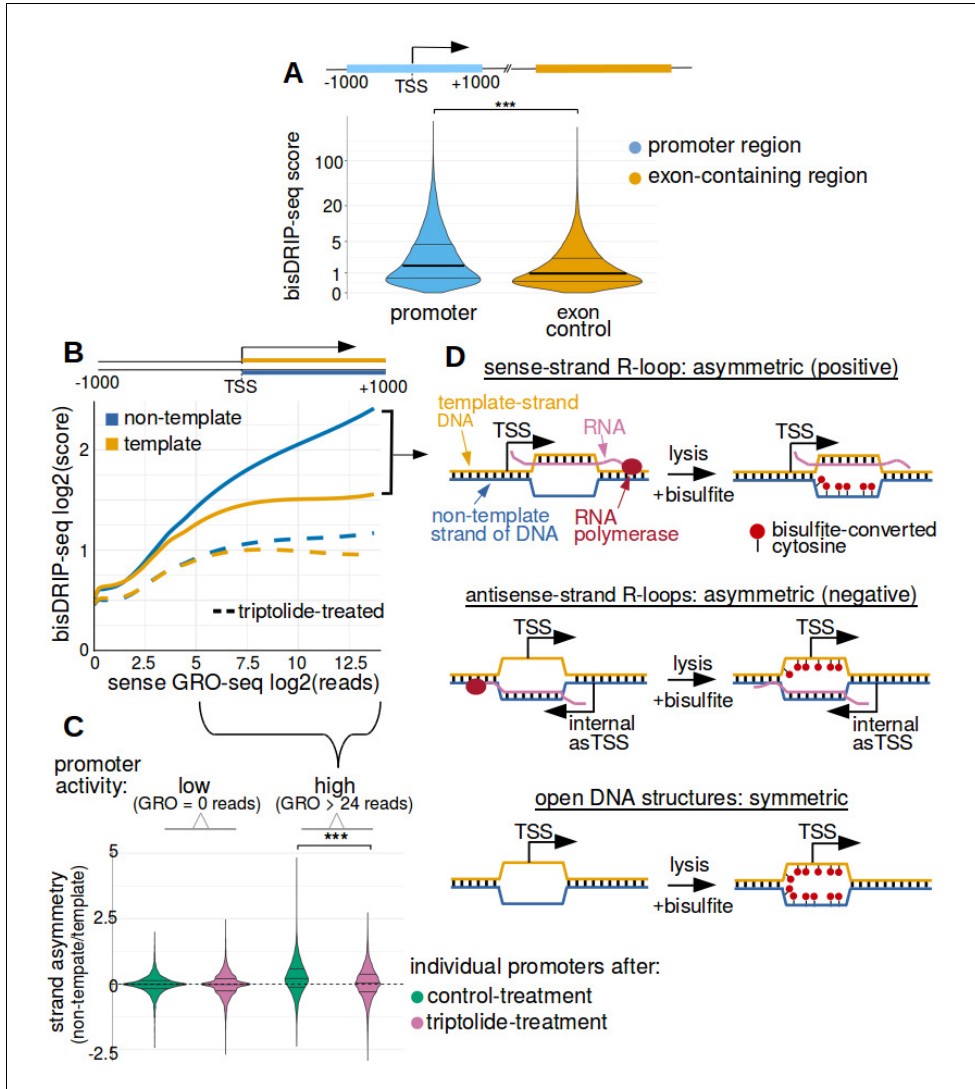

**Figure 2.** Transcription-dependent R-loops form in active promoter regions. (**A**) bisDRIP-seq scores in promoter regions tend to be higher than in matched exon-containing regions. R-loops were previously mapped to promoter regions (*Ginno et al., 2012*). To determine if bisDRIP-seq scores also map to promoter regions, we compared bisDRIP-seq scores in promoter regions with matched exonic regions. Promoter regions (blue) were defined as the region within a thousand base pairs of each transcription start site. For each promoter region, a matched region (orange) was selected downstream of the promoter region in the same gene centered on an exonic site chosen at random. The distribution of bisDRIP-seq scores (y-axis, mean bisDRIP-seq score from n = 13 samples) was plotted for promoter regions (blue, n = 60016) and matched regions (orange, n = 60016) using a violin plot. Within each violin plot, the fraction of genes with a given bisDRIP-seq score are represented by the width of the overlapping violin plot. Individual lines in the violin plot represent quartiles. bisDRIP-seq scores were significantly higher in promoter regions relative to matched exon-centered regions. 'TSS' refers to the transcription start site. The y-axis in the plot was log2 transformed. ***$p<2.2\times10^{-16}$, Wilcoxon signed-rank test. (**B**) R-loop formation correlates with promoter activity. Based on previous studies (*Sanz et al., 2016*), R-loops are expected to form in active promoter regions, rather than in inactive promoter regions. R-loops can be identified by bisDRIP-seq based on preferential labeling of the non-template strand. Therefore, we compared the bisDRIP-seq scores on the non-template strand to the scores on the template strand and determined whether this correlated with promoter activity. Promoter activity and bisDRIP-seq scores were assessed in the region between the transcription start site and + 1000 bp. bisDRIP-seq scores were assessed separately for the non-template (blue) and template (orange) strands. For each strand, a LOESS smoothed curve was plotted of the bisDRIP-seq scores (y-axis) at different levels of promoter activity (x-axis). This was repeated for control-treated samples (solid, mean bisDRIP-seq score from n = 13 samples) and samples treated with the transcription-inhibitor triptolide (dashed, mean bisDRIP-seq score from n = 2 samples). bisDRIP-seq scores on both strands are correlated with promoter activity. Notably, with increasing

*Figure 2 continued on next page*

*Figure 2 continued*

promoter activity, the non-template strand is preferentially labeled. This suggests that sense-strand R-loop form in these promoters. Promoter activity was assessed using a MCF-7 GRO-seq dataset from *Hah et al., 2013*. Both promoter activity and bisDRIP-seq scores were plotted on log2-transformed axes. Source data for figure included in *Figure 2—source data 1*. (C) Transcription-dependent R-loops form in active promoters. The presence of R-loops is suggested by bisDRIP-seq strand asymmetry as illustrated in *Figure 2D*. Here, strand asymmetry was calculated as the log2-fold ratio of the bisDRIP-seq score of the non-template strand relative to the bisDRIP-seq score of the template strand (y-axis). The distribution of strand asymmetry for promoter regions with high promoter activity (right, GRO-seq > 24 reads, n = 4895 promoter regions), as well as an equivalent number of inactive promoter regions (left, GRO-seq = 0 reads, n = 4895 promoter regions) was plotted using a violin plots. This was repeated for control-treated samples (green, mean bisDRIP-seq score from n = 13 samples) and triptolide-treated samples (pink, mean bisDRIP-seq score from n = 2 samples). Active promoter regions typically had higher non-template bisDRIP-seq scores than template-strand bisDRIP-seq scores in control-treated samples. This strand asymmetry was significantly reduced in triptolide-treated samples. These results suggest that there were transcription-dependent R-loops in active promoter regions. Promoter activity was assessed using a GRO-seq dataset from *Hah et al., 2013*. The width of violin plots represents the fraction of genes with the strand asymmetry plotted on the y-axis. The individual lines in violin plots represented quartiles. \*\*\*$p<2.2\times10^{-16}$, Wilcoxon signed-rank test. (D) Simple models of the structures that may explain the high bisDRIP-seq scores observed 3' of the transcription start site in *Figure 2B*. As illustrated, the sense-strand R-loops (top row) logically explain the strand 'positive' asymmetry observed in bisDRIP-seq scores. Additionally, the high bisDRIP-seq scores observed on both DNA strands of active promoters are likely explained by some combination of all three types of structure (all three rows). In these models, the vertical black hash marks between nucleic acid strands indicate that two strands are hybridized. Red circles refer to the location of bisulfite induced cytosine-to-uracil conversions. 'asTSS' refers to transcription start sites for antisense transcription. Source code for calculating bisDRIP-seq region scores can be found in *Source code 2*.

DOI: https://doi.org/10.7554/eLife.28306.007

The following source data and figure supplement are available for figure 2:

**Source data 1.** bisDRIP-seq scores of promoter regions.
DOI: https://doi.org/10.7554/eLife.28306.009

**Figure supplement 1.** Promoter regions were enriched in high bisDRIP-seq scores.
DOI: https://doi.org/10.7554/eLife.28306.008

Thus, bisDRIP-seq provides an approach to map R-loops at near-nucleotide resolution on a genome-wide scale.

## Near-Nucleotide resolution mapping of R-loops on a Genome-Wide scale

We used bisDRIP-seq to map single-stranded DNA throughout the genome. Thirteen bisDRIP-seq experiments were performed on different samples of MCF-7 cells. After performing bisDRIP-seq on these samples, the DNA fragments were sequenced using a traditional post-bisulfite library preparation method (see Materials and methods).

We next aligned the sequenced reads to the genome using Bismark, an alignment approach typically used for 5-methylcytosine mapping (*Krueger and Andrews, 2011*). This was necessary since the conversion of cytosines to uracils would confound traditional read alignment programs. Bismark was used to map conversions associated with single-stranded DNA as follows: first, reads were aligned to the genome. Then the cytosines that had been converted to uracils were identified (*Figure 1B*). As expected, reads were detected that contained only a single conversion, consistent with noise due to low-level double-stranded DNA cytosine conversions. However, reads and regions were also observed that contained consecutive cytosine conversions (*Figure 1—figure supplement 1A*). These reads are suggestive of single-stranded DNA.

We next applied our bioinformatic pipeline to distinguish the multiple conversions seen in single-stranded DNA regions from the stochastic conversions due to background noise.

In the first part of this pipeline, the stochastic rate of conversions was estimated. This was estimated based on the overall cytosine-to-uracil conversion rate in a given sample. Reads with a high percentage of conversions were excluded from this calculation since it was assumed that those conversions were not stochastic.

Next, we generated a 'bisDRIP-seq score' for each read. This score was calculated based on the number of cytosines converted in the read, relative to the likelihood of observing that number of conversions by stochastic noise (see Materials and methods for more details). Next, the score was normalized so that the sum of bisDRIP-seq scores within each sample was the same.

Next, we calculated bisDRIP-seq scores for individual nucleotide positions in the genome. The bisDRIP-seq score for each nucleotide position was calculated as the sum of the bisDRIP-seq scores of all of the reads that overlapped with that nucleotide position. These scores provide a read-length resolution map of single-stranded DNA that filters out stochastic conversions and that is more comparable between genomic regions (*Figure 1B*).

Importantly, bisDRIP-seq scores were substantially reduced at specific gene loci in samples treated with RNase H, which degrades RNA in RNA-DNA hybrids (*Figure 1—figure supplement 2*). This supports the idea that elevated bisDRIP-seq scores reflect R-loops.

We next performed two basic tests of the quality of data produced by bisDRIP-seq:

First, we asked whether reads with high bisDRIP-seq scores are randomly distributed across the genome or whether they are clustered in specific regions. To simulate a random distribution, Monte Carlo simulations were applied to our data. Relative to these Monte Carlo simulations, bisDRIP-seq scores were found to be clustered in specific regions (*Figure 1—figure supplement 1B–D*).

Second, we performed correlation tests between the bisDRIP-seq scores obtained in each of our samples. In all cases, there was significant correlation between bisDRIP-seq samples ($p < 10^{-16}$, Spearman's rank-correlation test, *Figure 1—figure supplement 3*).

## bisDRIP-seq scores show transcription-dependent enrichment in promoter regions

We next wanted to know if bisDRIP-seq scores are associated with promoter regions that contain R-loops. R-loops were previously mapped to promoter regions (*Ginno et al., 2012*), where they are thought to play important roles in gene expression (*Chen et al., 2015*). These R-loops, as mapped using DRIP-seq, were found to be transcription dependent and correlated with gene expression (*Sanz et al., 2016*). We therefore wanted to determine if bisDRIP-seq scores have a similar enrichment and transcription dependence in active promoter regions.

In order to investigate promoter regions, we compiled a list of transcription start sites using the GENCODE database (*Harrow et al., 2012*) (see Materials and methods). We then defined promoter regions, for the purpose of our analyses, as the region one kilobase on either side of each of these transcription start sites.

We next performed several simple analyses to ensure that the results from bisDRIP-seq experiments recapitulate the promoter-region enrichment observed in DRIP-seq studies. First, bisDRIP-seq scores, like DRIP-seq reads, were found to be enriched in promoter regions relative to downstream exon-containing regions (*Figure 2A* and *Figure 2—figure supplement 1A*). Next, we found that the number of DRIP-seq reads correlates with bisDRIP-seq scores in individual promoter regions ($p < 2.2 \times 10^{-16}$, Spearman's rank-correlation test, *Figure 2—figure supplement 1B*). Thus, bisDRIP-seq scores, like DRIP-seq reads, are enriched in promoter regions.

We next asked if bisDRIP-seq enrichment in promoter regions depends on active transcription. To test if bisDRIP-seq enrichment requires transcription, bisDRIP-seq was repeated using MCF-7 cells treated with the transcription-inhibitor triptolide (*Kupchan et al., 1972*; *Titov et al., 2011*; *Vispé et al., 2009*). bisDRIP-seq enrichment was reduced in these samples (*Figure 2—figure supplement 1C*). This suggests that bisDRIP-seq enrichment in promoter regions depends on ongoing transcription.

We next asked if the bisDRIP-seq scores in promoter regions are correlated with promoter activity. Promoter activity was assessed using existing GRO-seq datasets from MCF-7 cells cultured in a similar manner to our cells (*Hah et al., 2013*; *Hah et al., 2011*). GRO-seq measures the presence of active polymerases on a genome-wide scale (*Core et al., 2008*). Thus, these GRO-seq datasets allow us to distinguish between promoter regions with low and high promoter activity. We therefore compared the number of GRO-seq reads with bisDRIP-seq scores in promoter regions (*Figure 2—figure supplement 1D*). In this analysis, GRO-seq-measured promoter activity correlates with bisDRIP-seq scores ($p < 2.2 \times 10^{-16}$, Spearman's rank-correlation test). Thus, bisDRIP-seq scores are enriched in active promoter regions.

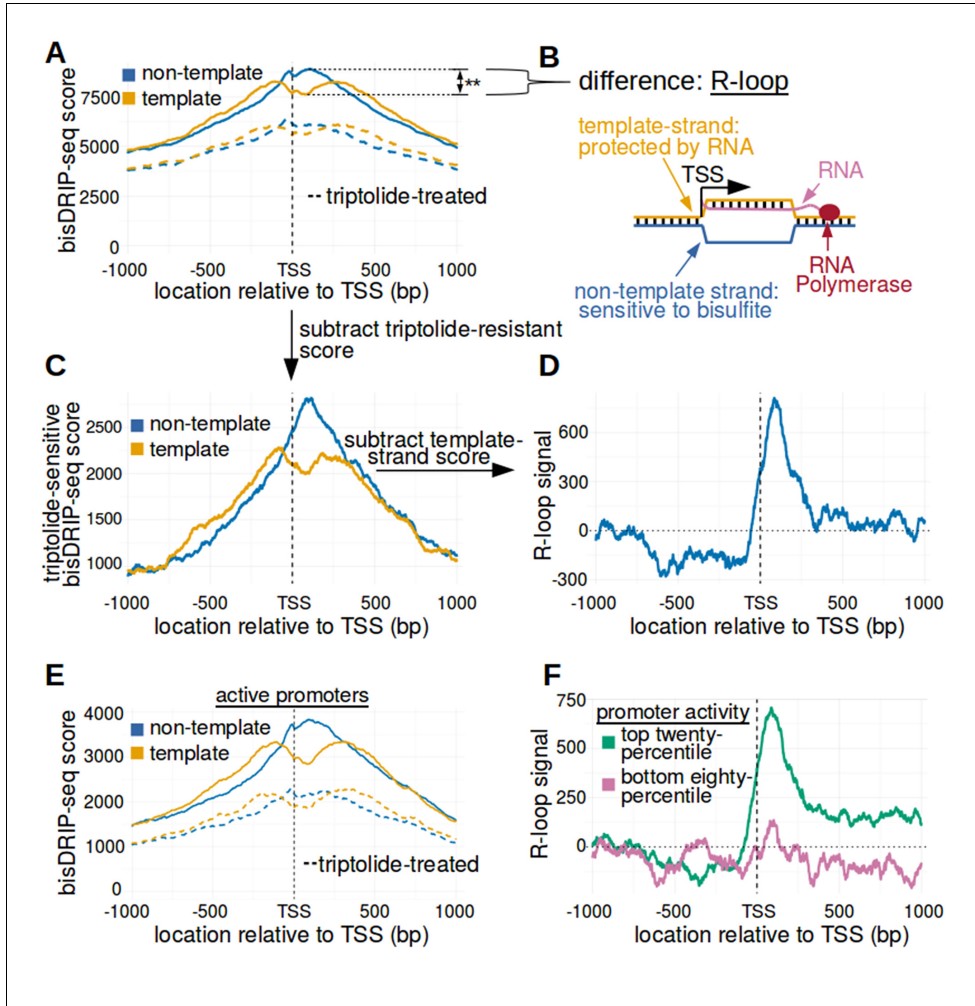

**Figure 3.** The 5' boundary of promoter-associated R-loops is located at the transcription start site. (**A**) Metaplot analysis of bisDRIP-seq scores reveals the location of promoter-associated R-loops. To determine the location of R-loops within promoter regions at read-length resolution, we used bisDRIP-seq score metaplot analysis. Metaplots were created by summing the bisDRIP-seq scores across all promoter regions (n = 78218) at each nucleotide position relative to the transcription start site. The score was calculated separately for the nucleotide position on the non-template strand (blue) and template strand (orange). Metaplots were then plotted for control-treated samples (solid, mean bisDRIP-seq score from n = 13 samples) and triptolide-treated samples (dashed, mean bisDRIP-seq score from n = 2 samples). In control-treated samples, bisDRIP-seq scores increase near the transcription start site for both strands. However, bisDRIP-seq scores were greater on the non-template strand than on the template strand immediately 3' of the transcription start site. This suggests that R-loops formed immediately 3' of the transcription start site. 'TSS' indicates the location of the transcription start site. **p<0.005, Wilcoxon signed-rank test. (**B**) Model of the sense-strand R-loops forming 3' of the transcription start site in *Figure 3A*. The location where strand asymmetry is observed in the bisDRIP-seq metaplot suggests that R-loops form 3' of the transcription start site. Additionally, the triptolide sensitivity of the bisDRIP-seq score asymmetry suggests that R-loops contain newly transcribed RNA. In this model, the black lines between nucleic acid strands indicate that two strands are hybridized. (**C**) Background subtraction more clearly reveals the location of R-loops. Triptolide-resistant bisDRIP-seq scores do not appear to reflect the presence of R-loops in *Figures 2B* and *3A*. As such, we repeated our metaplot analysis using triptolide-sensitive bisDRIP-seq scores. Metaplots of the template (orange) and non-template (blue) triptolide-sensitive scores (y-axis) in all promoter regions (n = 78218) were plotted relative to the transcription start site. The preferential labeling of the non-template strand immediately 3' of the transcription start site is more apparent in this plot. Triptolide-sensitive non-template bisDRIP-seq scores were generated by subtracting the triptolide-treated sample scores (mean bisDRIP-seq score from n = 2 samples) from the control-treated sample scores (mean bisDRIP-seq score from n = 13 samples). Triptolide-sensitive template bisDRIP-seq scores were generated in the same manner. (**D**) Metaplot of 'R-loop signal' reveals that R-loops form at the transcription start site. To better visualize the location of R-loops, we directly examined the

*Figure 3 continued on next page*

*Figure 3 continued*

difference in the bisDRIP-seq scores of the two strands by generating a metaplot of R-loop signal. R-loop signal was defined as the triptolide-sensitive template-strand bisDRIP-seq score subtracted from the triptolide-sensitive non-template bisDRIP-seq score. A metaplot of R-loop signal (y-axis) was plotted for all promoter regions (n = 78218) relative to the transcription start site. R-loop signal (blue) was highest in the region immediately 3' of the transcription start site and decreased 200–250 bp downstream of the transcription start site. R-loop signal at each nucleotide position was derived using the mean bisDRIP-seq score from n = 13 control-treated samples and mean bisDRIP-seq score from n = 2 triptolide-treated samples. (E) Metaplot of bisDRIP-seq scores in active promoter regions. In *Figure 2C,R*-loops appeared to predominantly form in active promoter regions. Thus metaplot analysis was repeated using only active promoter regions. A metaplot of bisDRIP-seq scores was created for the non-template strand (blue) and template strand (orange) across only active promoter regions (n = 15644). bisDRIP-seq scores were plotted on separate lines for control-treated samples (solid, mean bisDRIP-seq score from n = 13 samples) and triptolide-treated samples (dashed, mean bisDRIP-seq score from n = 2 samples). In control-treated samples, the location in the metaplot where the non-template bisDRIP-seq scores are higher than the template bisDRIP-seq scores is the same as in the metaplot for all promoters (*Figure 3A*). However, the difference between the strands is more pronounced in this metaplot of just active promoter regions. In this plot, active promoters refers to promoters with activity in the top twenty percentile as calculated using the GRO-seq dataset from *Hah et al., 2013*. (F) R-loops are only clearly observed in active promoter regions. A metaplot of R-loop signal was plotted for promoter regions in the top twenty percentile of promoter activity (green, n = 15644) and for promoter regions in the bottom eighty percentile of promoter activity (purple, n = 62574). There is no clear positive R-loop signal observed in promoter regions with lower promoter activity. By contrast, in active promoters the R-loop signal 3' of the transcription start site appears to be as high as the R-loop signal observed in all promoter regions (*Figure 3D*). R-loop signal at each nucleotide position was derived using the mean bisDRIP-seq score from n = 13 control-treated samples and mean bisDRIP-seq score from n = 2 triptolide-treated samples. Source code for metaplots can be found in *Source code 5*.

DOI: https://doi.org/10.7554/eLife.28306.010

The following figure supplements are available for figure 3:

**Figure supplement 1.** R-loop boundaries at the transcription start site of active promoter regions.
DOI: https://doi.org/10.7554/eLife.28306.011
**Figure supplement 2.** The transcription start site bounds R-loops in active promoter regions.
DOI: https://doi.org/10.7554/eLife.28306.012
**Figure supplement 3.** The strand asymmetry of bisDRIP-seq scores 3' of the transcription start site is sensitive to RNase H.
DOI: https://doi.org/10.7554/eLife.28306.013
**Figure supplement 4.** Antisense-strand R-loops were observed in promoter regions with high antisense-promoter activity.
DOI: https://doi.org/10.7554/eLife.28306.014

Notably, the enrichment of bisDRIP-seq scores in active promoter regions was reduced when RNA synthesis was blocked by triptolide treatment (*Figure 2—figure supplement 1D*).

Together, these results confirm that bisDRIP-seq scores, like DRIP-seq reads, are enriched in active promoter regions.

## Transcription-dependent R-loops form downstream of transcription start sites

We next wanted to determine if the bisDRIP-seq score enrichment in promoter regions is due to co-transcriptional R-loops. Conceivably, cytosine conversions could occur if single-stranded DNA is exposed as a result of other single-stranded DNA structures near transcription start sites, including unwound DNA due to supercoiling (*Hsieh and Wang, 1975*), G-quadruplexes (*Sen and Gilbert, 1988*), or genomic regions that contain paused polymerases (*Core et al., 2008*) that become more accessible to bisulfite after SDS treatment.

Relative to other single-stranded DNA structures, R-loops are known to produce a specific cytosine conversion signature upon bisulfite treatment. Specifically, cytosine conversions are limited to one strand of DNA in an R-loop. In other types of genomic structures that expose single-stranded DNA, cytosines on both strands of DNA can be converted. Thus efforts to map R-loops with bisulfite must demonstrate preferential labeling of one strand of DNA (*Yu et al., 2003*).

Cytosine conversions can also occur as a result of two types of R-loops: 'sense-strand R-loops' and 'antisense-strand R-loops.' Sense-strand R-loops refer to R-loops in which the RNA component of the R-loop is transcribed from the annotated promoter in the expected direction. In these R-loops, the non-template strand of DNA is exposed for bisulfite-mediated conversion. Antisense-strand R-loops, on the other hand, contain RNA that is transcribed from the opposite DNA strand. In this case, the template strand of DNA for the canonical gene transcript would be modified by bisulfite. Thus, the specific strand of DNA that exhibits bisulfite-mediated cytosine conversions indicates if a sense-strand R-loop or antisense-strand R-loop is present.

In order to distinguish between R-loops and other single-stranded DNA structures, we took advantage of the cytosine conversion strand asymmetry that is expected to result from R-loops. In other types of genomic structures that expose single-stranded DNA, cytosines on both strands of DNA can be converted.

To determine if the observed bisDRIP-seq enrichment downstream of the transcription start site in promoter regions is caused by sense-strand R-loops, we repeated our comparison of bisDRIP-seq score with promoter activity. However, here bisDRIP-seq scores were plotted separately for the template and non-template strands of DNA (*Figure 2B*). With increasing promoter activity we observed increasing bisDRIP-seq scores on both strands, with preferential labeling of the non-template strand of DNA. Thus, although both strands of DNA had bisulfite-induced cytosine conversions, we also observed asymmetric labeling of the non-template strand (*Figure 2C*). This is consistent with a mixture of single-stranded DNA structure and sense-strand R-loops in active promoter regions (*Figure 2D*).

Notably, the higher bisDRIP-seq scores on the non-template strand than the template strand were largely eliminated in triptolide-treated samples (*Figure 2B and C*). This suggests that the sense-strand R-loops in the promoter region are transcription dependent.

## The transcription start site is the 5' boundary of promoter-associated R-loops

The exact starting position and ending positions of promoter-associated R-loops remains unclear. This is due to the low resolution of conventional R-loop mapping methods (*Chen et al., 2015*; *Ginno et al., 2012*). We wanted to take advantage of the high resolution of bisDRIP-seq to map the exact boundaries of R-loops in promoter regions.

We first asked where R-loops are located in relation to transcription start sites. In many promoters, transcription initiates from multiple nearby transcription start sites (*Carninci et al., 2006*). This creates a practical limit to the precision that we can achieve in mapping the location of R-loops relative to transcription start sites.

We first mapped R-loops relative to all transcription start sites using metaplots of bisDRIP-seq scores. First, bisDRIP-seq scores were calculated for each nucleotide position surrounding the transcription start site in all individual promoter regions. Promoter regions were defined using the GENCODE database described above. Then, the bisDRIP-seq score at a given nucleotide position relative to the transcription start site was summed across all promoter regions. These scores were then plotted separately for the non-template strand and the template strand (*Figure 3A*).

The resulting metaplot suggests that a mix of R-loops and other single-stranded structures surround the transcription start site. The presence of single-stranded DNA at transcription start sites is suggested by the peak of bisDRIP-seq scores near the transcription start site (*Figure 3B*). The location of these single-stranded structures is consistent with previous maps of single-stranded DNA (*Kouzine et al., 2013*). However, the presence of R-loops is specifically suggested by asymmetric, preferential labeling of the non-template strand (*Figure 3B*). Indeed, the non-template strand bisDRIP-seq scores are significantly higher than the template strand bisDRIP-seq scores immediately 3' of the transcription start site (p<0.005, Wilcoxon signed-rank test) (*Figure 3A*, *Figure 3—figure supplement 1B*). This suggests that the transcription start site is the 5' boundary of promoter-associated R-loops.

We next asked whether the R-loops bounded by the transcription start site are dependent on transcription. We repeated our metaplot analysis using the samples treated with triptolide. In these samples, there was minimal difference between the template and non-template strand bisDRIP-seq scores (*Figure 3A*). The small remaining difference in scores upon triptolide-treatment may reflect a real difference, but it may also reflect noise in our measurements. Overall, the loss of bisDRIP-seq

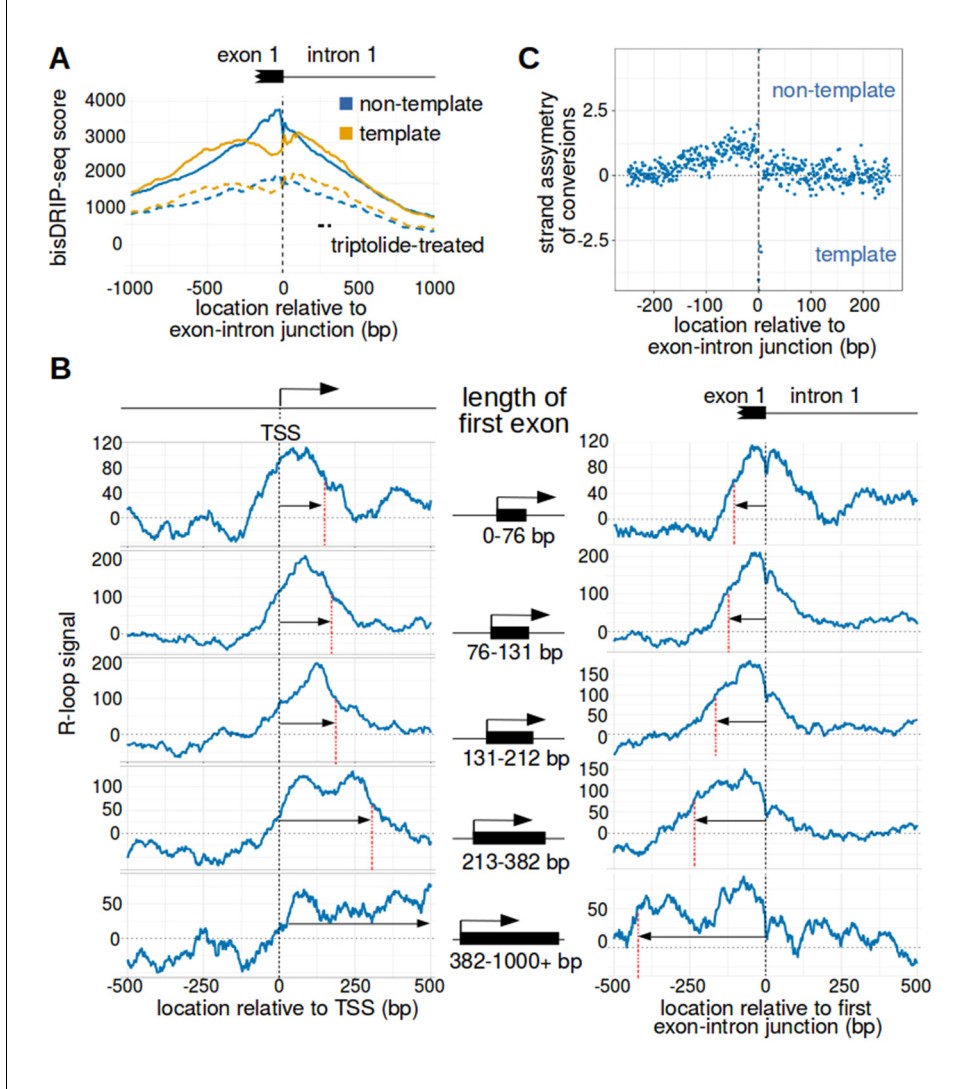

**Figure 4.** The first exon-intron junction appears to act as a 3' boundary to R-loops. (**A**) Strand asymmetry in bisDRIP-seq scores ends at the first exon-intron junction. To determine where R-loops are located relative to the first exon-intron junction, a metaplot of bisDRIP-seq scores was generated relative to the first exon-intron junction. bisDRIP-seq scores were calculated for control-treated samples (solid, mean bisDRIP-seq score from n = 13 samples) and triptolide-treated samples (dashed, mean bisDRIP-seq score from n = 2 samples). These mean bisDRIP-seq scores were then summed at each position relative to the first exon-intron junction for all intron-containing gene promoter regions (n = 14538). These values were plotted separately for the template strand (orange) and for the non-template strand (blue). Immediately 5' of the exon-intron junction, the non-template bisDRIP-seq scores were greater than the template bisDRIP-seq scores in control-treated samples. This difference in bisDRIP-seq scores between the two strands was eliminated almost immediately 3' of the exon-intron junction. This suggests that the first exon-intron junction acted as a 3' boundary to promoter-associated R-loops. (**B**) R-loop-forming regions expand further from both the transcription start site and the exon-intron junction in promoter regions with larger first exons. To test if the first exon-intron junction was acting as a boundary to R-loops, we created metaplots of R-loop signal in bins of promoter regions with different first-exon sizes. Promoter regions were binned into five groups based on the size of their first exon (n = 2907 or 2908 per bin). On the left side of the panel, metaplots of R-loop signal (blue) centered on the transcription start site were plotted for each bin of promoter regions. A vertical, dashed line (red) indicates the 3'-most location where R-loop signal was at half of the maximum signal in the metaplot. The arrow pointing to the dashed line indicates the distance from the line to the transcription start site. In the middle of the panel are schematics indicating the range of first-exon sizes observed in the bin of promoter regions that is examined in the adjacent metaplots. As illustrated, the smallest first exons are examined in the bin displayed in the top row and each subsequent row examines a bin containing larger first

*Figure 4 continued on next page*

*Figure 4 continued*

exons. On the right side of the panel, metaplots of R-loop signal (blue) centered on the first exon-intron junction were plotted for each bin of promoter regions. A vertical, dashed line (red) indicates the 5'-most location where R-loop signal was at half of the maximum signal in the metaplot. The arrow pointing to the dashed line indicates the distance from the line to the transcription start site. In metaplots representing genes with longer first exons, R-loop signal extended further 3' from the transcription start site and further 5' of the first exon-intron junction. These results suggest that R-loops were typically bounded between the transcription start site and the first exon-intron junction. 'TSS' refers to the transcription start site. In each case, R-loop signal at each nucleotide position was derived using the mean bisDRIP-seq score from n = 13 control-treated samples and mean bisDRIP-seq score from n = 2 triptolide-treated samples. (C) The 3' boundary of promoter-associated R-loops is within a few base pairs of the first exon-intron junction. To determine, at near-nucleotide resolution, the 3' R-loop boundary relative to the first exon-intron junction, we generated a metaplot of bisDRIP-seq-conversion asymmetry. To generate this metaplot, the total number of conversions was summed at each position relative to the first exon-intron junction for all intron-containing gene promoter regions (n = 14538). This was performed separately for each strand of the control-treated sample (mean of n = 13 samples) and then these values were background-corrected by subtracting the same values from the triptolide-treated samples (mean of n = 2 samples). Finally, the strand asymmetry of conversions was calculated as the log ratio of the number of conversions on the non-template strand relative to the template strand. This strand asymmetry of conversions (y-axis) was plotted for each position relative to the exon-intron junction (x-axis). In this metaplot, there is asymmetry in the strand orientation of conversions immediately 5' of the exon-intron junction, with more conversions on the non-template strand. However, within a few base pairs 3' of the exon-intron junction, conversions appear to be equally distributed on both the template and non-template strand. The consensus splice site confounds this analysis to some extent at the exact splice site. Nevertheless, this analysis suggests that the 3' R-loop boundary is located within base pairs of the first exon-intron junction. See *Figure 4—source data 1* for source data regarding the set of exon-intron junctions studied in *Figure 4A and B*.

DOI: https://doi.org/10.7554/eLife.28306.015

The following source data and figure supplement are available for figure 4:

**Source data 1.** Location of exon-intron junctions analyzed in *Figure 4*.

DOI: https://doi.org/10.7554/eLife.28306.017

**Figure supplement 1.** The first exon-intron junction is the only intron-exon junction that is observed to act as a boundary for R-loop formation.

DOI: https://doi.org/10.7554/eLife.28306.016

score strand asymmetry upon triptolide treatment demonstrates that the enrichment of R-loops 3' of the transcription start site requires transcription.

The transcription start site boundary was even more apparent after applying a background correction. To do this, we defined an 'R-loop signal,' which reflects the difference in bisDRIP-seq labeling of the non-template strand from the template strand after using the triptolide-treated samples for background correction. Thus, the template strand metaplot from the triptolide-treated sample was subtracted from the template strand metaplot from the control sample. The same background correction was used for the non-template strand metaplot. This background correction further enhances the demarcation of the transcription start site as the 5' boundary of promoter-associated R-loops (*Figure 3C*). Next, we subtracted the template strand metaplot from the non-template metaplot to generate a metaplot of the R-loop signal (*Figure 3D*). In this plot, the 5' boundary of R-loop signal at the transcription start site is very pronounced.

We repeated this analysis on promoters that have either high promoter activity or low promoter activity (*Figure 3E and F*, *Figure 3—figure supplement 1C*, *Figure 3—figure supplement 2*). A peak of R-loop signal was only observed immediately 3' of the transcription start site in the analysis of active promoters. This is consistent with our previous analysis showing that R-loops specifically form 3' of the transcription start site in active promoters.

Notably, the observed difference in strand bisDRIP-seq scores 3' of the transcription start site in active promoters is lost after RNase H treatment (*Figure 3—figure supplement 3*). This confirms that the R-loop signal observed in active promoter regions is caused by R-loops.

Taken together, these data indicate that the transcription start site demarcates the 5' boundary of promoter-associated R-loops. Additionally, since only noise was observed from promoter regions with low promoter activity, these regions were removed from future analysis unless otherwise noted.

## The first exon-intron junction acts as a 3' boundary to promoter-associated R-loops

We next wanted to know if there is a 3' boundary to R-loops. Based on the metaplot analysis in *Figure 3D*, R-loop signal drops approximately 200–250 bp downstream of the transcription start site. This is further from the transcription start site than the typical first post-transcription start site nucleosome (*Schones et al., 2008*) and the location of promoter-proximal RNA polymerase II pausing (*Core et al., 2008*), suggesting that these features probably do not impede R-loop expansion. On the other hand, 200–250 bp is reasonably close to the median distance between the transcription start site and the first exon-intron junction, which is 181 bp in our dataset (see Materials and methods). Also, previous studies found that knockdown of the 5' splice site-binding factor SRSF1 induces the formation of R-loops (*Li and Manley, 2005*), which could be explained if splicing is involved in bounding R-loop expansion. These pieces of evidence suggested that the first exon-intron junction might act as the 3' boundary to R-loop expansion in promoter regions.

We therefore asked where R-loops are located relative to the first exon-intron junction. We used the 5' end of the first intron as the reference point for a metaplot of bisDRIP-seq scores. Intronless genes were not considered in this analysis. In these metaplots, we observed that the bisDRIP-seq scores on the non-template strand are significantly higher than on the template strand immediately 5' of the first exon-intron junction (*Figure 4A* and *Figure 4—figure supplement 1A*). This strand asymmetry in bisDRIP-seq scores drops 3' of the exon-intron junction. This suggests that the 3' end of R-loops are bounded by the first exon-intron junction.

We further tested the idea that the first exon-intron junction is the 3' boundary of R-loops. To do this, we asked if the size of the R-loop-forming region in promoter regions increases with the length of the first exon. To test this question, metaplot analysis was repeated on groups of promoter regions with different sized first exons. First, promoter regions were binned into five groups based on the annotated size of the first exon. Then, metaplots were created of the R-loop signal centered around either the transcription start site or the first exon-intron junction (*Figure 4B*). Strikingly, in groups of promoter regions with longer first exons, the R-loop signals are also longer. This supports the idea that R-loops are bounded at both the transcription start site and the first exon-intron junction.

We next asked if we could map the location of the 3' R-loop boundary at near-nucleotide resolution. bisDRIP-seq scores are shared across entire reads, which limits the resolution of using bisDRIP-seq scores to read length resolution. On the other hand, cytosine-to-uracil conversions should map R-loops at near-nucleotide resolution. We therefore generated a metaplot of the strand asymmetry of cytosine conversions relative to the first exon-intron junction. First, the number of conversions on either strand were background corrected by subtracting the number of conversions observed in our triptolide bisDRIP-seq data. Next, the log ratio of conversions on the non-template strand relative to the template strand was plotted at each site relative to the exon-intron junction (*Figure 4C*). In this metaplot there is a decrease in the relative number of conversions on the non-template strand within base pairs of the exon-intron junction (*Figure 4—figure supplement 1B*). Thus, it appears that the R-loop boundary is within a few base pairs of the first exon-intron junction.

We next asked whether other exon-intron junctions also act as R-loop boundaries. In particular, we focused on the junctions between the first intron and the second exon, as well as the junction between the second exon and the second intron. We repeated our metaplot analysis by plotting R-loop signal at each position relative to the given exon-intron or intron-exon junction (*Figure 4—figure supplement 1A*). No clear peak in R-loop signal is observed near these downstream intron-exon junctions. This suggests that only the first exon-intron junction acts as a boundary to R-loop formation.

Together, these results suggest that there is a boundary to R-loop formation located within base pairs of the first exon-intron junction.

## bisDRIP-seq reveals evidence for antisense-strand R-loops

We noticed that there is negative R-loop signal upstream of the transcription start site (*Figure 3D and F*). This could be caused by 'antisense-strand R-loops,' i.e., with antisense RNA transcripts hybridized to the annotated non-template strand of DNA (See *Figure 3—figure supplement 4C* for this structure). Antisense-strand R-loops would result in more prominent bisulfite conversions on the

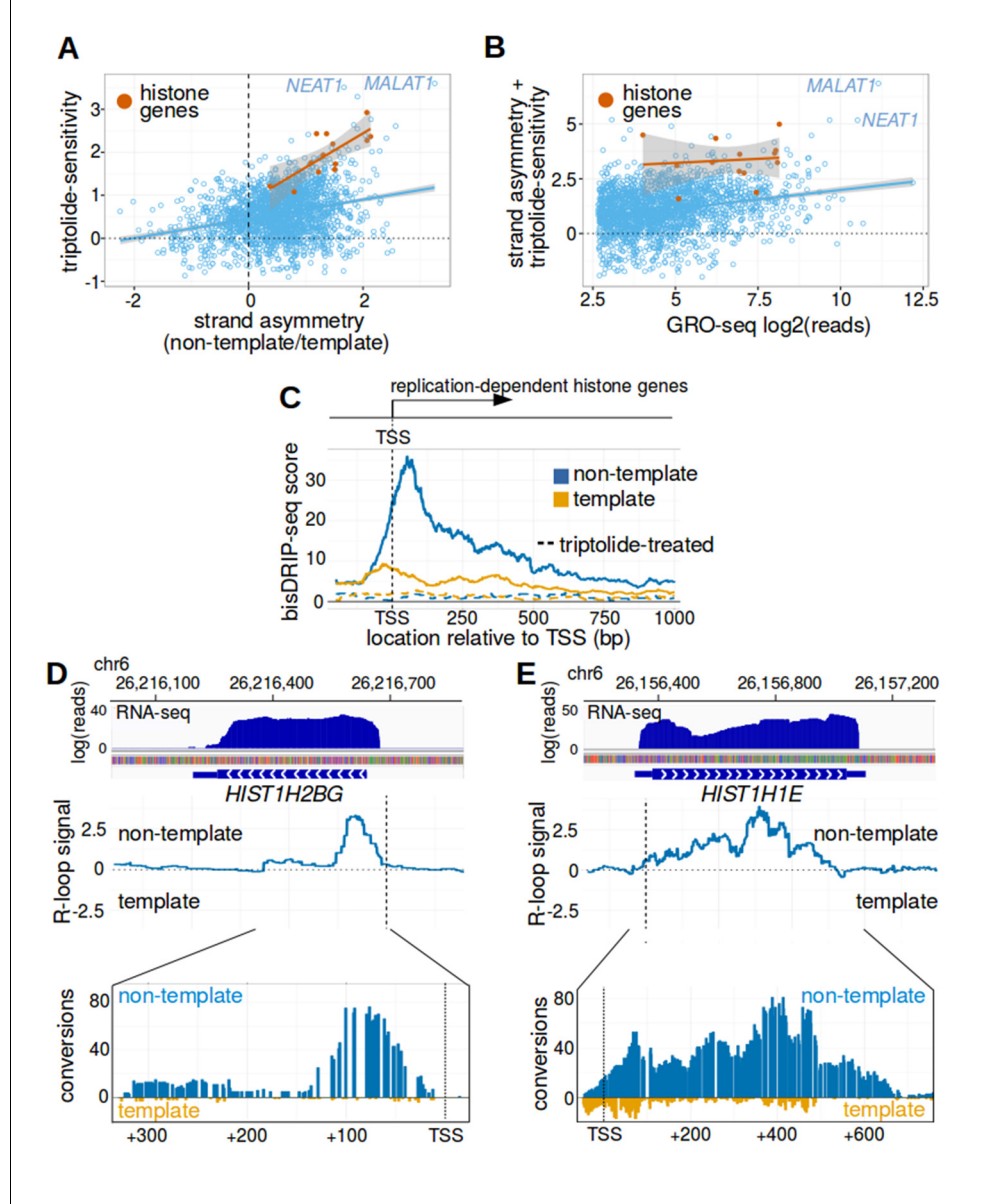

**Figure 5.** Replication-dependent histone genes frequently form R-loops. (**A**) Genes that are strongly associated with promoter-associated R-loops were identified using two criteria. First, transcription-dependent R-loop formation was identified based on the triptolide-sensitivity of the promoter region bisDRIP-seq score (x-axis). Second, R-loop formation was identified by the strand asymmetry of the promoter region bisDRIP-seq score (y-axis). Promoter regions for all genes except those encoding replication-dependent histones (n = 2064, blue) were plotted on these two axes alongside a linear regression model (blue line). Promoter regions for replication-dependent histones (n = 13, orange) were plotted alongside a separate linear regression model (orange line). In the resulting plot, there appears to be a group of genes with relatively high scores on both axes. This suggests that these genes are strongly associated with transcription-dependent R-loops formation. This set of genes included the replication-dependent histones genes and the indicated lncRNA genes *MALAT1* and *NEAT1*. In this plot, all bisDRIP-seq score measurements were made in the region between the transcription start site and + 250 bp. Triptolide-sensitivity was calculated as the non-template bisDRIP-seq score in triptolide treated-samples (mean bisDRIP-seq score from n = 2 samples) subtracted from the non-template bisDRIP-seq score from control-treated samples (mean bisDRIP-seq score from n = 13 samples). Strand asymmetry was calculated as the log2 ratio of the non-template bisDRIP-seq score to the template bisDRIP-seq score in control-treated samples (mean bisDRIP-seq score from n = 13 samples). The shaded areas around both linear regression models represent 95% confidence intervals. (**B**) Promoter activity does not explain the association between histone genes and R-loop formation.

*Figure 5 continued on next page*

*Figure 5 continued*

R-loop formation was calculated for each promoter region by taking the sum of the x-axis and y-axis values from *Figure 5A*. R-loop formation was then plotted against promoter activity for each promoter region. Promoter regions for non-histone genes (n = 2064, blue) were plotted alongside a linear regression model (blue) for these genes. Promoter regions for replication-dependent histone genes (n = 13, orange) were plotted alongside a separate linear regression model (orange). Promoter activity was correlated with R-loop formation. Nonetheless, all of the examined replication-dependent histone genes have higher R-loop formation scores than most of the genes with similar promoter activity. Promoter activity (x-axis) was calculated between the transcription start site and +250 bp using the GRO-seq dataset from *Hah et al., 2013*. R-loop formation was calculated as the sum of bisDRIP-seq score strand asymmetry and triptolide-sensitivity. These values were derived from the mean bisDRIP-seq scores of n = 13 control-treated samples and mean bisDRIP-seq scores of n = 2 triptolide-treated samples. The shaded areas around both linear regression models represent 95% confidence intervals. (C) R-loops are prevalent in the promoter regions of replication-dependent histone genes. The location of R-loops in the promoter regions of replication-dependent histone genes was examined using metaplot analysis. A metaplot was generated of bisDRIP-seq scores relative to the transcription start sites of all replication-dependent histone genes (n = 69). bisDRIP-seq scores were plotted separately for the template (orange) and non-template (blue) strands. This was repeated for control-treated samples (solid, mean bisDRIP-seq score from n = 13 samples) and triptolide-treated samples (dashed, mean bisDRIP-seq score from n = 2 samples). The bisDRIP-seq scores on the non-template strand were higher than the scores on the template strand throughout the promoter region 3′ of the transcription start site. Additionally, almost no signal was observed in the triptolide-treated samples. These results suggest that R-loops are the predominant structure observed by bisDRIP-seq in the promoter regions of this class of genes. 'TSS' indicates the location of the transcription start site. (D,E) Gene-specific heterogeneity in the 3′ R-loop boundaries of histone genes. To examine whether promoter-associated R-loops have 3′ boundaries in intronless genes, R-loop signal was examined in select histone genes. The observed heterogeneity in 3′ R-loop boundaries is represented by (E) *HIST1H2BG* and (F) *HIST1H1E*. At the top of each panel are the gene loci and the sense-strand RNA-seq reads that map to the loci containing either gene. In the middle of each panel is the R-loop signal (y-axis) plotted across the genomic loci containing each gene (blue line). In the lower plot, cytosine-to-uracil conversions were mapped to the genomic loci containing each gene. The number of conversions on the template strand and non-template strand were plotted below the x-axis (orange) or above the x-axis (blue), respectively. Shown are the total number of conversions observed in all bisDRIP-seq samples (n = 13). The sharp drop in R-loop signal near the start of the *HIST1H2BG* gene contrasts with the relatively stable R-loop signal observed in *HIST1H1E*. Transcription start sites are indicated by both 'TSS' and by dashed vertical lines. R-loop signal at each nucleotide position was derived using the mean bisDRIP-seq score from n = 13 control-treated samples and mean bisDRIP-seq score from n = 2 triptolide-treated samples. See *Figure 5—source data 1* for source data for *Figure 5A and B*.

DOI: https://doi.org/10.7554/eLife.28306.018

The following source data and figure supplements are available for figure 5:

**Source data 1.** R-loop scoring of promoters studied in *Figure 5A, B*.
DOI: https://doi.org/10.7554/eLife.28306.022

**Figure supplement 1.** Histone genes are enriched in R-loops.
DOI: https://doi.org/10.7554/eLife.28306.019

**Figure supplement 2.** R-loop boundaries in a subset of replication-dependent histone genes.
DOI: https://doi.org/10.7554/eLife.28306.020

**Figure supplement 3.** Conclusions of bisDRIP-seq results are not altered by normalizing bisDRIP-seq scores to account for the number of cytosines per read.
DOI: https://doi.org/10.7554/eLife.28306.021

---

annotated template strand. This type of labeling is opposite from the non-template strand labeling that is caused by the predominant type of R-loop that forms from sense transcription.

We considered that antisense transcription could lead to antisense-strand R-loops that generate these negative R-loop signals. To test this, we calculated the antisense-transcription activity of each promoter region upstream of the transcription start site using the previously described GRO-seq dataset (*Hah et al., 2013*). In promoter regions with high antisense-transcription promoter activity, the template-strand bisDRIP-seq scores upstream of the transcription start site were significantly higher than on the non-template strand (*Figure 3—figure supplement 4A and B*). These data indicate a correlation between antisense transcription and antisense R-loops.

We next used the promoters that showed the highest level of antisense-transcription promoter activity to generate a metaplot of R-loop signal. In this metaplot, the negative R-loop signal was prominent upstream of the transcription start site (*Figure 3—figure supplement 4D*). Taken together, these data suggest that some promoter regions contain antisense-strand R-loops and that this is linked to antisense transcription in these promoter regions.

## R-loops are observed in the promoter regions of intronless genes

We next wanted to identify the promoters that show the strongest association with transcription-dependent R-loops. Any unique features associated with these promoters may be directly related to the R-loops forming at these promoter regions. We searched for promoter regions with two major features: First, we searched for promoter regions that showed disproportionately high bisDRIP-seq score on the non-template strand compared to the template strand. Second, we searched for promoter regions where the majority of the bisDRIP-seq score on the non-template strand was lost upon triptolide treatment. In this analysis, we noticed a set of promoter regions that exhibited both of these features (*Figure 5A*). We therefore ranked genes based on the sum of these two features to identify the genes that show the strongest association with R-loop structures (*Table 1*).

Two important classes of genes were identified in this analysis and both lack introns. First, six of the top 25 genes (24%) and nine of the top 50 genes (18%) are replication-dependent histone genes. The second class of genes encode intronless noncoding RNAs, including *MALAT1*, *NEAT1*, *RPPH1*,

**Table 1.** The 25 genes that were most strongly associated with transcription-dependent sense-strand R-loop structures in the region between the transcription start site and + 250 bp.

| Gene | Histone | Single exon | Genetype |
|---|---|---|---|
| *MALAT1* | no | yes | lncRNA |
| *CTB-58E17.1* | no | yes | lncRNA |
| *RHOB* | no | yes | protein coding |
| *NEAT1* | no | yes | lncRNA |
| *HIST1H2BC* | yes | yes | protein coding |
| *STX16* | no | no | protein coding |
| *ARFIP2* | no | no | protein coding |
| *HIST1H2BG* | yes | yes | protein coding |
| *XBP1* | no | no | protein coding |
| *TM7SF2* | no | no | protein coding |
| *DNAJB1* | no | no | protein coding |
| *HIST1H2BK* | yes | yes | protein coding |
| *MIEN1* | no | no | protein coding |
| *NFKBIA* | no | no | protein coding |
| *RP11-166B2.1* | no | no | protein coding |
| *SRSF3* | no | no | protein coding |
| *LASP1* | no | no | protein coding |
| *RPPH1* | no | yes | ribozyme |
| *HIST4H4* | yes | yes | protein coding |
| *RPL23* | no | no | protein coding |
| *HIST1H2BD* | yes | yes | protein coding |
| *GSS* | no | no | protein coding |
| *HIST1H1E* | yes | yes | protein coding |
| *SRSF7* | no | no | protein coding |
| *TPM3* | no | no | protein coding |

DOI: https://doi.org/10.7554/eLife.28306.023

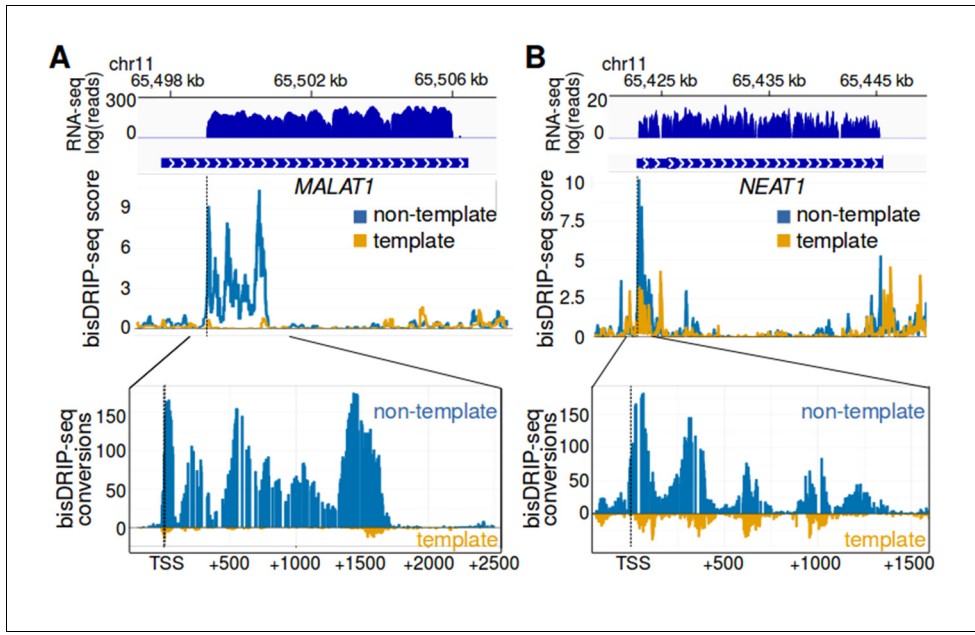

**Figure 6.** *MALAT1* and *NEAT1* contain large, bounded promoter-associated R-loops. (**A**) The promoter-associated R-loop forming region in *MALAT1* is large, but bounded. *MALAT1* had the strongest association with R-loops in *Figure 5A* and it is a longer intronless gene than the previously examined replication-dependent histone genes. To determine how far the R-loop-forming region in *MALAT1* extends into the gene body, the bisDRIP-seq signal at the *MALAT1* locus was examined. At the top of the panel is the gene model of *MALAT1* and the sense-strand RNA-seq reads that mapped to this region from an ENCODE MCF-7 RNA-seq dataset (blue, plotted on a log axis). Under the gene model of *MALAT1*, bisDRIP-seq scores were mapped to the genomic region containing *MALAT1* (mean bisDRIP-seq score from n = 13 samples). The bisDRIP-seq score on the template strand (orange) and non-template strand (blue) were plotted separately. In the lower plot, cytosine-to-uracil conversions were mapped to the genomic region surrounding the *MALAT1* R-loop forming region. The number of conversions on the template strand and non-template strand were plotted below the x-axis (orange) or above the x-axis (blue), respectively. Shown are the total number of conversions observed in all bisDRIP-seq samples (n = 13). The *MALAT1* R-loop region extended from the transcription start site to a site approximately 1750 base pairs downstream of the transcription start site. The transcription start site, indicated by 'TSS' and a dashed vertical line, was determined based on the MCF-7 ENCODE CAGE-seq dataset ENCFF207DXM and was located at ch11:65,499,042. (**B**) The promoter-associated R-loop forming region in *NEAT1* appears to have reduced R-loop further into the *NEAT1* gene body. R-loop formation in the *NEAT1* locus was examined since *NEAT1* is adjacent to *MALAT1* and was also strongly associated with R-loop formation. At the top of the panel is a gene model of *NEAT1* and the sense-strand RNA-seq reads that mapped to this region from an ENCODE MCF-7 RNA-seq dataset (blue, plotted on a log axis). Below the gene model of *NEAT1*, the bisDRIP-seq scores were mapped to the genomic region containing *NEAT1* (mean bisDRIP-seq score from n = 13 samples). The bisDRIP-seq score on the template strand (orange) and non-template strand (blue) were plotted separately. In the lower plot, cytosine-to-uracil conversions were mapped to the genomic region surrounding the *NEAT1* R-loop forming region. The number of conversions on the template strand and non-template strand were plotted below the x-axis (orange) or above the x-axis (blue), respectively. Shown are the total number of conversions observed in all bisDRIP-seq samples (n = 13). The *NEAT1* R-loop forming region extended almost 1500 base pairs from the transcription start site. However, R-loop signal showed periodicity and appeared to decrease gradually from the transcription start site to its final 3' boundary. The gene model of *NEAT1* represents the 23 kb *NEAT1* isoform.

DOI: https://doi.org/10.7554/eLife.28306.024

The following figure supplements are available for figure 6:

**Figure supplement 1.** R-loop formation in *MALAT1* and *NEAT1*.

DOI: https://doi.org/10.7554/eLife.28306.025

**Figure supplement 2.** R-loop signal in *MALAT1* and *NEAT1* is reduced by RNase H treatment.

DOI: https://doi.org/10.7554/eLife.28306.026

**Figure supplement 3.** Analysis of individual reads suggests that there is heterogeneity in the R-loop structures forming in the *MALAT1* locus.

DOI: https://doi.org/10.7554/eLife.28306.027

*Figure 6 continued*

**Figure supplement 4.** The R-loop in *MALAT1* ends at an SRSF1 binding site.
DOI: https://doi.org/10.7554/eLife.28306.028

and *CTB-58E17.1*. We noticed that two other top hits, the protein-coding genes *RHOB* and *JUNB*, are also intronless genes.

In general, intronless genes appear to be enriched among the promoters that show strong association with R-loop structures. In total, 44% of the top 25 genes in our list are intronless, compared to approximately 2% of long non-coding RNAs (lncRNAs) and 3% of protein-coding genes (*Derrien et al., 2012*; *Louhichi et al., 2011*). The presence of R-loops in the promoter regions of these intronless genes suggests that the first exon-intron junction does not promote R-loop formation.

## Replication-dependent histone genes are strongly associated with R-loops

The presence of R-loops in the replication-dependent histone genes is potentially interesting given how these genes are regulated. As their name suggests, these histone genes are regulated in a cell-cycle dependent manner (*Robbins and Borun, 1967*). They also lack poly-A tails and are processed in special histone bodies in the nucleus (*Dominski and Marzluff, 1999*). Thus these genes are co-regulated using special processing pathways.

We first considered the possibility that the prominent R-loop signal detected in histone promoter regions simply reflects high promoter activity in these genes. However, histone genes consistently had higher R-loop-associated bisDRIP-seq scores than the majority of genes with similar promoter activity (*Figure 5B*). This suggests that promoter activity does not explain the strong R-loop signal observed in histone genes.

Similar analysis indicated that nuclear RNA levels and recruitment of RNA polymerase II also do not explain the R-loop signal observed in histone genes (*Figure 5—figure supplement 1A and B*).

We next examined the prevalence of R-loops and other single-stranded DNA structures in the promoter regions of histone genes. We repeated our metaplot analysis using the bisDRIP-seq scores from the entire class of replication-dependent histone genes (*Figure 5C*). In the resulting metaplot, bisDRIP-seq scores are low on the template strand and in triptolide-treated samples. This suggests that there is little single-stranded structure outside of R-loops in the promoter regions of these genes. In contrast, bisDRIP-seq scores are high downstream of the transcription start site on the non-template strand. Moreover, these high bisDRIP-seq scores are not observed after RNase H treatment (*Figure 5—figure supplement 1C–E*). Together, these results indicate that histone genes, as a group, have a high level of R-loop formation relative to the formation of other single-stranded DNA structures.

We next asked whether R-loops are bounded in replication-dependent histone genes. It was not clear if R-loops would be bounded in these genes since they lack introns and therefore they lack exon-intron junctions (*Marzluff et al., 2008*). Conceivably, the R-loops could extend to the entire length of the transcript. We therefore identified the boundaries of the R-loop signal in each of the nine histone genes that had the highest propensity to form R-loops (*Figure 5A*). As expected, the 5' boundary of the R-loops appear to be near the transcription start site in all nine genes. In five of the nine histone genes, the entire R-loop appeared to be restricted to the initial portion of the gene (*Figure 5D* and *Figure 5—figure supplement 2A–E*). Sequence analysis of these boundaries does not reveal a clear sequence enrichment or motif (*Figure 5—figure supplement 1D and E*), making it currently unclear how this boundary is determined. In other cases, like *HIST1H1E*, R-loops seemed to cover nearly the entire gene (*Figure 5E* and *Figure 5—figure supplement 2F–I*). Together these results suggest that additional factors may establish 3' R-loop boundaries in a subset of the replication-dependent histone genes.

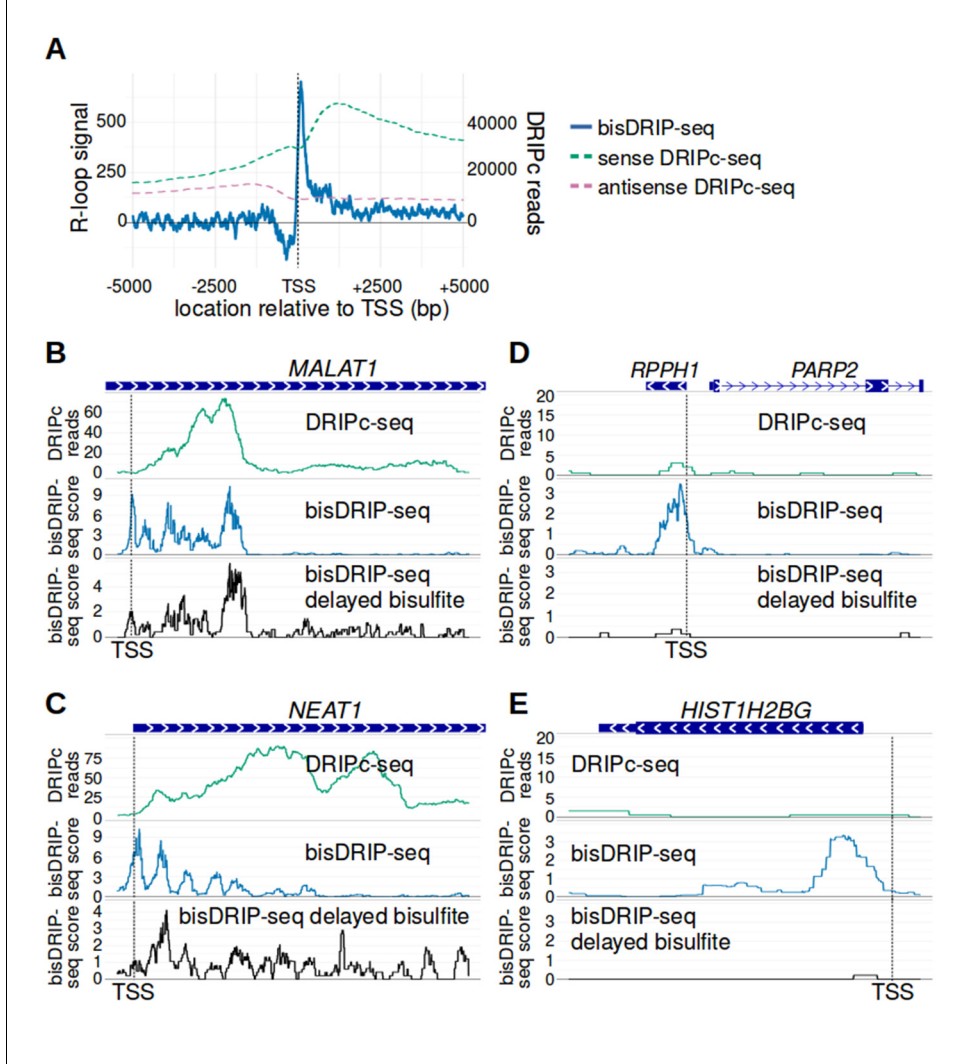

**Figure 7.** bisDRIP-seq provides improved mapping of R-loops relative to DRIPc-seq. (**A**) The well-defined R-loop boundaries observed by bisDRIP-seq are less well demarcated in DRIPc-seq. To determine if the resolution of bisDRIP-seq R-loop mapping improves upon the resolution of previous R-loop mapping methods, we compared the data from bisDRIP-seq and DRIPc-seq. DRIPc-seq maps the location of RNAs that are incorporated into R-loops and, apart from bisDRIP-seq, it is the highest resolution method to map R-loops (*Sanz et al., 2016*). A metaplot of bisDRIP-seq R-loop signal (y-axis, solid blue) was plotted for active promoter regions (n = 15644) relative to the transcription start site. Overlapped onto the bisDRIP-seq metaplot are metaplots of sense-strand and antisense-strand DRIPc-seq reads. R-loop signal (blue) was highest in the region immediately 3' of the transcription start site and decreased 200–250 bp downstream of the transcription start site. Although there is also a peak of sense-strand DRIPc-seq reads 3' of the transcrption start site, there is no sharp 3' boundary to these reads within 5000 bp of the transcription start site. This suggests that bisDRIP-seq can map R-loop boundaries at a higher resolution than existing R-loop mapping methods. R-loop signal was defined as the triptolide-sensitive template-strand bisDRIP-seq score subtracted from the triptolide-sensitive non-template bisDRIP-seq score. This R-loop signal at each nucleotide position was derived using the mean bisDRIP-seq score from n = 13 control-treated samples and mean bisDRIP-seq score from n = 2 triptolide-treated samples. DRIPc-seq reads reflect the mean number of reads at a given site from n = 2 DRIPc-seq experiments calculated using the datasets from *Sanz et al., 2016*. The location of the transcription start site is demarcated by 'TSS' and a dashed vertical line. (**B**–**E**) A subset of R-loops are unstable ex vivo. The structure of R-loops has previously been shown to change in vitro (*Kaback et al., 1979*; *Landgraf et al., 1996*). To determine whether the structure of R-loops isolated from cells can change ex vivo, we compared R-loop maps from experiments in which R-loop mapping occurred either during cell lysis (bisDRIP-seq) or hours after cell lysis (delayed bisDRIP-seq and DRIPc-seq). In the delayed bisDRIP-seq experiment (lower plot), R-loops were labeled with bisulfite 16 hr after cell lysis (black, y-axis, n = 1 sample) rather

*Figure 7 continued*

than during cell lysis. R-loops were mapped to the gene loci of (**B**) *MALAT1,* (**C**) *NEAT1,* (**D**) *RPPH1* and (**E**) *HIST1H2BG* using DRIPc-seq (top plot, green, mean of n = 2 samples), bisDRIP-seq (middle plot, blue, n = 13 samples) and delayed bisDRIP-seq (lower plot, black, n = 1 sample). For (**A**) *MALAT1* and (**B**) *NEAT1*, R-loops map to similar regions in all three experiments. However, the R-loop signal appears to migrate towards the 3' end of the gene in both genes. On the other hand, the R-loop signal observed for (**C**) *RPPH1* and (**D**) *HIST1H2BG* in control-treated bisDRIP-seq samples is not observed in either the delayed bisDRIP-seq or DRIPc-seq datasets. Together, these results suggest that a subset of R-loops are unstable ex vivo. DRIPc-seq data was obtained from *Sanz et al., 2016*.

DOI: https://doi.org/10.7554/eLife.28306.029

## Large R-loops form immediately downstream of the transcription start site in *MALAT1* and *NEAT1*

Another set of genes which preferentially exhibit R-loops in their promoter regions are *MALAT1* and *NEAT1*. *MALAT1* and *NEAT1* are adjacent genes that encode abundant, intronless lncRNAs (*Hutchinson et al., 2007*). These lncRNAs remain in the nucleus where they are involved in the regulation of transcription (*Hirose et al., 2014*) and splicing (*Tripathi et al., 2010*), respectively. Both *MALAT1* and *NEAT1* are longer than 3 kb, which is longer than the replication-dependent histone genes studied above. We were therefore interested in whether there are boundaries to R-loop expansion in these much longer intronless genes.

We first asked where R-loops are located in *MALAT1* and *NEAT1*. The R-loop forming region in *MALAT1* extends from the transcription start site to a position approximately 1700 bp downstream, with a sharp decrease in R-loop signal downstream of this position (*Figure 6A*). Similarly, the R-loop in *NEAT1* extended approximately 1400 bp from the transcription start site (*Figure 6B*). Beyond this site, there was minimal detectable R-loop signal. As with the R-loops in the replication-dependent histone genes, these R-loops showed nearly complete loss of bisDRIP-seq signal on the non-template strand after triptolide treatment (*Figure 6—figure supplement 1A and B*). Moreover, the high bisDRIP-seq scores in this region are not observed after RNase H treatment (*Figure 6—figure supplement 2A and B*). This suggests that relatively long R-loops form in *MALAT1* and *NEAT1* and that these R-loops are bounded to the 5' end of each gene.

Although our mapping reveals the location of the R-loops in *MALAT1* and *NEAT1* at near-nucleotide resolution, analysis of previous DRIP-seq datasets (*Sanz et al., 2016*) reveal signals in the same overall regions (*Figure 6—figure supplement 1A and B*).

Interestingly, there appears to also be periodicity in the bisDRIP-seq scores on the non-template strand of *NEAT1* with peaks and valleys every 300 base pairs (*Figure 6B*). This phenomenon is also detectable, but less prominent in *MALAT1* (*Figure 6A*).

These valleys may indicate the existence of smaller R-loops in some *MALAT1* or *NEAT1* genes in some cells. This idea is supported by examining individual reads within the R-loop forming region in *MALAT1*. In most cases, we observed that cytosines were almost completely converted in individual reads (*Figure 6—figure supplement 3A*). However, we also observed a subset of reads with long stretches of cytosine conversions on one end and long stretches of unconverted cytosines on the other end (*Figure 6—figure supplement 3B*). The region where the conversions stop occurring in individual reads might reflect an internal border of a R-loop. It should be noted that we cannot exclude the possibility that this may reflect the location of a structured region in the single-stranded DNA that prevents bisulfite reactivity. Nevertheless, the presence of peaks and valleys within the R-loop-forming region of *NEAT1* and *MALAT1* raises the possibility of heterogeneity in the size and location of the individual R-loops within the larger R-loop forming region identified by bisDRIP-seq.

## Comparison of bisDRIP-seq to existing R-loop mapping methods

Previous efforts to map R-loops have primarily relied on immunoprecipitation of RNA-DNA hybrids (*Chédin, 2016*). Traditionally, DNA fragments containing an R-loop are recovered and sequenced. The sequenced fragments contain both the DNA involved in the R-loop and regions of DNA that are not in the R-loop. Newer approaches, like DRIPc-seq (*Chen et al., 2015*; *Sanz et al., 2016*), can provide higher resolution by sequencing the RNA component of the R-loop. We therefore next wanted

to determine if the specific R-loop boundaries detected by bisDRIP-seq could also be detected in the human DRIPc-seq datasets.

We first compared the location of R-loop signal in metaplots of DRIPc-seq and bisDRIP-seq signal in active promoters regions. As we demonstrated earlier, the R-loop signal in bisDRIP-seq is bounded between the transcription start site and the first exon-intron junction (*Figure 7A*). Thus, we observe a tight peak of R-loop signal within a few hundred base pairs of the transcription start site in active promoter regions. On the other hand, DRIPc-seq shows a marked enrichment of reads following the transcription start site; however, there is no clear boundary of sense-strand DRIPc-seq reads peaks within 5 kb of the transcription start site (*Figure 7A*). This highlights the improvement in resolution obtained by bisDRIP-seq.

We next compared the R-loop maps generated using DRIPc-seq or bisDRIP-seq at individual gene loci. In some genes, such as *MALAT1* and *NEAT1,* we observed tight concordance between the R-loop maps generated from DRIPc-seq and bisDRIP-seq (*Figure 7B and C*). However, the boundaries demarcated by bisDRIP-seq appear more clear than the boundaries demarcated by DRIPc-seq. On the other hand, there were few DRIPc-seq reads mapped to genes such as *RPPH1* and *HIST1H2BG*, which displayed strong R-loop signal in bisDRIP-seq (*Figure 7D and E*). This suggests the possibility that bisDRIP-seq might identify a set of R-loops that are not observable by DRIP-seq.

We next asked if the size or location of the R-loop might change ex vivo. bisDRIP-seq labels R-loops upon cell lysis, while other mapping methods recover the R-loop hours or days after cell lysis. R-loops could conceivably, expand, contract, or disappear during this time period. This seemed plausible, since the structure of some R-loops has previously been observed to change in solution (*Kaback et al., 1979*; *Landgraf et al., 1996*). To test this possibility, we performed a 'delayed-bisulfite' bisDRIP-seq experiment. In this experiment, we delayed the bisulfite step of our protocol so that samples were only treated with bisulfite 16 hr after cell lysis. In some genes, such as *MALAT1* and *NEAT1*, the bisDRIP-seq scores were similar when using the delayed-bisulfite protocol. In other cases, the delayed-bisulfite treatment was associated with a marked depletion of the R-loop signal in *RPPH1* and *HIST1H2BG* (*Figure 7B–E*). This suggests the possibility that some R-loops may not be stable for prolonged periods of time ex vivo and therefore may not be observed in approaches that do not label or recover R-loops during lysis.

## Discussion

The precise location of R-loops in promoter regions has been obscure due to the low resolution of conventional R-loop mapping approaches (*Chen et al., 2015*; *Ginno et al., 2012*). These approaches rely on the S9.6 antibody to recover and sequence genomic fragments containing R-loops. Here we describe bisDRIP-seq, a near-nucleotide resolution method for mapping R-loops. In this approach, single-stranded DNA regions are identified based on their reactivity with bisulfite. These regions were then mapped throughout the genome based on the location of bisulfite-induced cytosine-to-uracil conversions. In some regions, bisulfite-induced cytosine conversions are enriched on one strand of DNA, show a requirement for RNA transcription and are removed following RNase H treatment, supporting the idea that these regions contain R-loops. R-loops were previously thought to expand thousands of base pairs into gene bodies without any clear boundary to their expansion or formation (*Chédin, 2016*). Here, we discover boundaries to R-loop formation at the transcription start site and the first exon-intron junction. The discovery of these boundaries suggest that the maximum length of most promoter-associated R-loops is predetermined by the exon-intron structure of genes.

Identification of R-loops by bisDRIP-seq depends primarily on bisulfite labeling of the single-stranded component of R-loops. In addition to bisulfite labeling, bisDRIP-seq takes advantage of the S9.6 antibody to enrich for R-loop containing fragments of DNA, thus leading to enhanced read coverage in R-loop-containing regions. However, S9.6 is known to react to other nucleic acid structures, including structured RNA (*Phillips et al., 2013*), and its full specificity relative to genomic DNA has not been examined. In our analysis, non-R-loop promoter single-stranded structures were immunoprecipitated by S9.6 antibody. This may reflect the poor specificity of the S9.6 antibody. Therefore, simply sequencing DNA recovered by S9.6 may not provide sufficient specificity for R-loop mapping.

bisDRIP-seq overcomes problems with the S9.6 antibody by adding multiple criteria to selectively identify which recovered fragments contain R-loops. These criteria included preferential labeling of a single strand of DNA and the requirement that the labeling be transcription-dependent. S9.6 antibody enrichment in protocols like DRIP-seq and bisDRIP-seq have been shown to be biased towards promoter regions (*Halász et al., 2017*). The requirement for preferential bisulfite labeling of one strand of DNA should largely mitigate this bias. Still, it is worth nothing that these criteria may have resulted in us excluding specific genomic regions that form RNA-DNA hybrids on both DNA strands or that contain extremely stable R-loops. Nevertheless, these criteria allow us to exclude non-R-loop single-stranded structures that might otherwise be mistaken for R-loops.

The location of the 5' boundary of R-loops, identified here to be located at the transcription start site, suggests that R-loops are primarily formed using the canonical gene transcript. Conceivably, other transcripts could form R-loops in promoter regions, such as promoter upstream transcripts (*Preker et al., 2011*) or extracoding RNAs (*Savell et al., 2016*). Our data suggests that other transcripts initiating upstream or downstream of the canonical transcription start site are less likely to be the major source of RNA in promoter-associated R-loops.

Why would the exon-intron junction serve as the 3' R-loop boundary? The simplest explanation is that RNA splicing limits the length of the hybrid that can form between the spliced RNA and the template strand of DNA. If the RNA that forms the R-loop is spliced, then the RNA would not be able to hybridize to intron-encoding DNA (*Figure 8A*). Alternatively, R-loop expansion might be blocked by proteins that are bound to the 5' splice site in the RNA (*Figure 8B*). Both of these mechanisms could potentially explain the exon-intron junction R-loop boundary.

Previous studies found that the relative location of the first exon-intron junction impacts the rate of transcription (*Bieberstein et al., 2012*; *Brinster et al., 1988*; *Fong and Zhou, 2001*). Smaller first exons are associated with higher rates of transcription and enhanced recruitment of transcription initiation factors (*Bieberstein et al., 2012*). Conceivably, if smaller first exons are associated with less efficient or less stable R-loops, this may account for the higher transcription associated with these genes.

Although intronless genes lack exon-intron junctions, we identified R-loops and R-loop boundaries in these genes. Intronless genes were among the genes with the highest R-loop signal immediately downstream of the transcription start site. In particular, two classes of intronless genes were identified: intronless lncRNAs, including *MALAT1* and *NEAT1*, and replication-dependent histones.

Histone genes are notable for their unusual replication-dependent transcription regulation and for their non-canonical end termination (*Marzluff et al., 2008*). During DNA synthesis, histone genes are transcribed and processed more efficiently than at other parts of the cell-cycle (*Hereford et al., 1981*). Additionally, the histone genes are the only mRNAs that lack poly-A tails. Like *MALAT1* and *NEAT1*, the histone genes are terminated with a special stem-loop structure (*Marzluff et al., 2008*). This special processing has been previously linked to transcription and the absence of histone gene splicing. It is intriguing to speculate that either of these processes may relate to the robust R-loop structures that are seen at the 5' end of these genes.

*MALAT1* and *NEAT1*, which are both intronless genes, also had two of the strongest R-loop signals. These genes clearly show that promoter-associated R-loops can be bounded to specific regions of genes in the absence of an exon-intron junction. RNA-binding proteins might determine R-loop boundaries in these types of intronless genes. Indeed, we observed prominent binding of SRSF1 at a position that corresponds to the 3' boundary of the R-loop in *MALAT1* in publicly-available datasets (*Figure 6—figure supplement 4*). Thus, RNA-protein complexes may also limit the expansion of R-loops, especially in intronless genes.

Interestingly, all of the intronless genes that we studied are components of nuclear bodies. The histone RNAs, *MALAT1* and *NEAT1* associate with histone locus bodies, nuclear speckles and paraspeckles, respectively (*Clemson et al., 2009*; *Hutchinson et al., 2007*; *Liu et al., 2006*). The strong R-loop signal in so many genes with roles in well-established nuclear bodies raises the possibility that R-loops may play a role in the formation or regulation of these bodies. In the case of at least the histone genes and *NEAT1*, formation of the nuclear bodies is known to occur near the DNA encoding these RNA transcripts (*Clemson et al., 2009*; *Mao et al., 2011*). This suggests the possibility that R-loop formation may be involved in tethering RNA to chromatin to facilitate ribonucleoprotein assembly. Future research will be needed to test this and other possibilities for why these nuclear body-associated genes are prone to R-loop formation.

Although bisDRIP-seq was used here to study steady-state promoter-associated R-loops, the method can be used for studying other questions regarding R-loops. For example, this method could be used to study the dynamic changes in R-loop size and location in response to various signals like estrogen. Moreover, as transcription termination sites become better annotated, high-resolution R-loop could guide the study of R-loops involved in transcription termination.

# Materials and methods

## Key resources table

| Reagent type (species) or resource | Designation | Source or reference | Identifiers | Additional information |
|---|---|---|---|---|
| cell line (human) | MCF-7 cells | ATCC | HTB-22 | |
| antibody | S9.6 antibody | Kerafast | ENH001 | Used at a concentration of 20 mg/L |
| commercial assay or kit | Pico-Methyl Seq Library Prep Kit | Zymo Research | D5455 | |
| chemical compound, drug | Ammonium sulfite monohydrate | Sigma-Aldrich | 358983–500G | |
| chemical compound, drug | 45% ammonium bisulfite | Pfaltz and Bauer Inc | A29946250g | |
| chemical compound, drug | sodium bisulfite | Sigma-Aldrich | 243973–100G | |
| chemical compound, drug | triptolide | R and D Systems | 3253 | |
| software, algorithm | Bismark | Babraham Bioinformatics | version 0.14.3 | |
| software, algorithm | Flexbar | *Dodt et al., 2012* PMID: 24832523 | version 2.5 | |

## Cell culture

MCF-7 cells were used for all experiments. Cell lines were originally obtained from ATCC. Cell identification was performed by ATCC, which identifies cell lines using STR profiling, and cell identity was regularly checked by visual inspection of morphologies. Independently prepared vials of cell lines were tested for mycoplasma by Hoechst staining; these cell lines were tested after the conduction of the experiments described here.

MCF-7 cells (10 million per 15 cm tissue culture dish) were cultured for three days at 37°C in 50 ml of Gibco's phenol red-free Dulbecco's Modified Eagle's Medium (Thermo Fisher Scientific, Waltham, Massachusetts) containing 5% charcoal-stripped fetal bovine serum (Thermo Fisher Scientific, Waltham, Massachusetts). Media was replaced after two days. In these experiments, we included MCF-7 cells treated for different amounts of time with 100 µM estrogen (Sigma-Aldrich, Saint Louis, Missouri) prior to lysis to ensure transcriptional activation at a broad set of genes (*Hah et al., 2011*). The use of estrogen in some samples ensures that we can maximally capture R-loops that occur in either basal and stimulated conditions.

Each of the thirteen control-treated bisDRIP-seq experiments and each of the two triptolide-treated bisDRIP-seq experiments were performed using a separate dish of MCF-7 cells. Thus, each bisDRIP-seq replicate used a different sample of MCF-7 cells that had, at a minimum, been cultured on a separate dish for three days.

## bisDRIP-seq protocol

In the bisDRIP-seq protocol, R-loops are treated with bisulfite during cell lysis under non-denaturing conditions. The basic concept underlying this approach is that bisulfite can interact with single-stranded DNA, but cannot interact with DNA in a double helix (*Yu et al., 2003*). This contrasts with 5-methylcytosine mapping, which is performed under denaturing conditions that cause all double-stranded DNA to be single stranded and susceptible to bisulfite treatment. It is therefore worth noting that bisDRIP-seq could be affected by 5-methylcytosine. If 5-methylcytosine is present in the single-stranded DNA of the R-loop, there will be minimal conversion at those sites. This should not substantially impact R-loop mapping, since 5-methylcytosines are found only in a CpG sequence

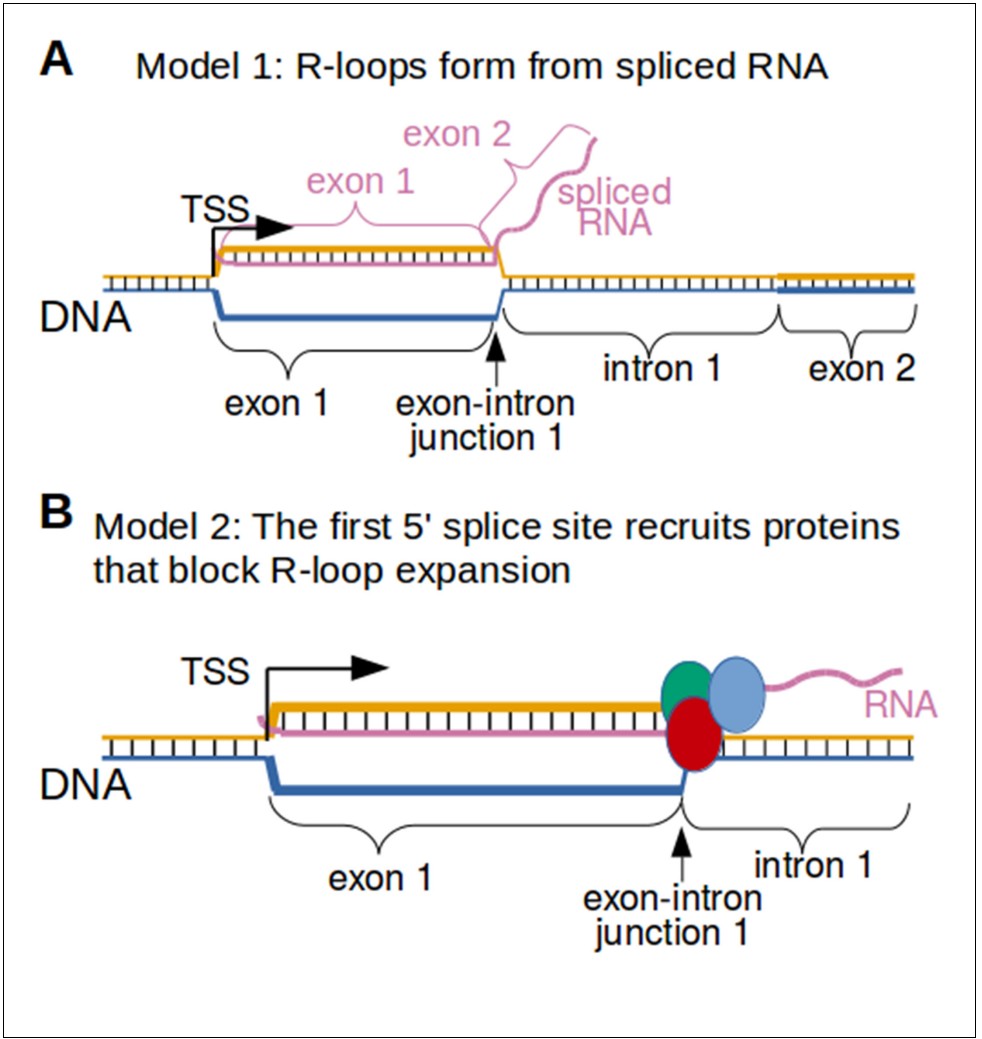

**Figure 8.** Two possible models to explain the R-loop boundaries observed in the promoter regions of intron-containing genes. (**A**) Spliced RNA R-loop model. Promoter-associated R-loops may be bounded by the exon-intron junction because the RNA component of R-loops is spliced. If the first intron is spliced out of the RNA that is incorporated into promoter-associated R-loops, then the RNA cannot hybridize to the region of DNA encoding the first intron. This would prevent R-loops from expanding past exons. In this model, the template strand for canonical transcription is represented in orange, while the opposite, non-template strand is represented in blue. Black lines between two nucleic acid strands indicate that the nucleic acids are hybridized. 'TSS' refers to the transcription start site. (**B**) Protein-recruitment model. Alternatively, the first exon-intron junction may recruit proteins that prevent R-loop formation in the intron region. For example, the first exon-intron junction recruits splicesome proteins, which have been implicated in R-loop processing (*Li and Manley, 2005*). These or other proteins may bind the exon-intron junction and then block R-loop expansion. In this model, the template strand for canonical transcription is represented in orange, while the opposite, non-template strand is represented in blue. The red, green and blue circles represent proteins associated with first exon-intron junction. Black lines between two nucleic acid strands indicate that the nucleic acids are hybridized.
DOI: https://doi.org/10.7554/eLife.28306.030

context and most R-loops will have some cytosines that are not followed by guanine. Nevertheless, this issue could be considered during data analysis.

Prior to performing bisDRIP-seq, lysis buffer was prepared containing 55.5 mg of ammonium sulfite monohydrate (Sigma-Aldrich, Saint Louis, Missouri), 760 µl of 45% ammonium bisulfite (Pfaltz and Bauer Inc, Waterbury, Connecticut), 85.5 µl of 20 mM hydroquinone (Sigma-Aldrich, Saint Louis,

Missouri) and 62.7 µl of 0.5 M ethylenediaminetetraacetic acid (EDTA) in a final volume of 2.85 ml. Lysis buffer was prepared using degassed water.

Additionally, ammonium bisulfite solution was prepared containing 0.67 g ammonium sulfite monohydrate, 2.08 g of sodium bisulfite (Sigma-Aldrich, Saint Louis, Missouri), 5 ml of 45% ammonium bisulfite in a final volume of 6 ml. This ammonium bisulfite solution was prepared using degassed water, following the methodology described by *Hayatsu et al., 2004*.

Immediately prior to cell lysis, media was removed from cells. Cells were washed three times in ice-cold PBS. Cells were then scraped off of plates into ice-cold PBS. These cells were then transferred to a 15 ml conical tube. The conical tube containing cells was centrifuged for 3 min at 300 g . PBS was then removed from cells.

Next, cells were lysed in the presence of bisulfite. Bisulfite was included in the lysis buffer to achieve rapid single-stranded DNA modification that would minimize R-loop expansion or contraction. First, lysis buffer (1.5 ml) was added to the cells. Cells in lysis buffer (475 µl) were added to each of two microcentrifuge tubes, each of which contained 25 µl of lithium dodecyl sulfate. Samples were then incubated in a shaking incubator for 30 min at 37°C at 1100 rpm, while being protected from light.

Next, the concentration of bisulfite in samples was increased in order to increase the rate of the irreversible second hydrolytic deamination step in the bisulfite conversion process (*Hayatsu et al., 2004*). The concentration of bisulfite was increased by transferring samples to a microcentrifuge tube containing 1 ml of ammonium bisulfite solution and 26.8 µl of 20 mM hydroquinone. Samples were then incubated in a shaking incubator for 2 hr at 37°C at 1100 rpm, while being protected from light.

Next, bisulfite was removed from the samples through dialysis. First, samples were added to 2.5 ml of dialysis buffer (50 mM Tris pH 8, 10 mM EDTA) in a 15 ml conical centrifuge tube. Samples were then added to an Amicon Ultra-4 Centrifugal Filter Unit with a 30,000 nominal molecular weight limit (EMD Millipore, Darmstadt, Germany). Next, the samples were centrifuged following the instructions of the centrifugal filter unit manufacturer. After centrifugation, fresh dialysis buffer was added to the centrifugal filter unit and centrifugation was repeated. This dialysis process was repeated until the sample was nominally dialyzed at least 200 fold. Proteinase K buffer (50 mM NaCl, 50 mM Tris pH 8, 10 mM EDTA) was then added to the sample and centrifugation was repeated. This dialysis process was repeated until the sample was nominally dialyzed at least 5000 fold in Proteinase K buffer. At this point, the samples were resuspended in 1 ml of Proteinase K buffer and transferred to a microcentrifuge tube.

Next, we used Proteinase K (Life Technologies, Carlsbad, California) to degrade the proteins in our samples. In order to treat our samples with Proteinase K, 50 µl of 20% SDS and 200 µl of 20 µg/µl Proteinase K was added to each sample. Samples were then incubated overnight at 37°C while being rotated end over end.

Nucleic acids were purified from any remaining proteins present in the sample using phenol-chloroform. Samples were split into two microcentrifuge tubes. An equal volume of Phenol-chloroform (Life Technologies, Carlsbad, California) was added to each sample. Samples were mixed and then centrifuged for five minutes at 18400 g. The aqueous phase of each sample was transferred to a new microcentrifuge tube. Each sample was then combined with an equal volume of chloroform. Samples were mixed and centrifuged for two minutes at 18400 g. The aqueous phase of each sample was transferred to a new microcentrifuge tube. Samples were then ethanol precipitated for 2 hr at 4°C. The precipitated DNA was then transferred and combined in a new tube by swirling the DNA around a pipette tip. Finally, the DNA was washed with 70% ethanol.

Next, the DNA was fragmented using restriction enzymes. The precipitated DNA was resuspended in 850 µl of NEB buffer 3.1 (100 mM NaCl, 50 mM Tris-HCl pH 8.0, 10 mM MgCl$_2$, 100 µg/ml bovine serum albumin). The samples were then digested overnight at 37°C by a cocktail of restriction enzymes, which included: 200 U/ml *Hind*III, 200 U/ml *Eco*RI, 100 U/ml *Bsr*GI, 200 U/ml *Xba*I and 50 U/ml *Ssp*I (all from New England Biolabs, Ipswich, Massachusetts) in a total volume of 900 µl.

Next, the quality of the DNA digest was assessed and the concentration of DNA was measured in preparation for the RNA-DNA hybrid immunoprecipitation. DNA digest efficiency was measured by running the DNA on a 0.8% agarose gel stained with ethidium bromide. This was performed for every bisDRIP-seq experiment and samples were only used for immunoprecipitation if the digested DNA was mostly between 1 kb and 10 kb long. The concentration of DNA was measured using a

Quant-iT dsDNA Assay Kit (Thermo Fisher Scientific, Waltham, Massachusetts) and a SpectraMax M2 Microplate Reader (Molecular Devices, Sunnyvale, California) following the instructions provided by the Quant-iT dsDNA Assay kit manufacturers.

Next, RNA-DNA hybrids were enriched using an immunoprecipitation protocol modeled after the DRIP-seq methodology described by *Ginno et al. (2012)*. Digested DNA (20 µg) was added to 20 µg of S9.6 antibody (Kerafast, Boston, Massachusetts) in 1 ml of immunoprecipitation buffer (10 mM sodium phosphate pH 7.0, 140 mM NaCl, 0.05% Triton X-100). The sample was then incubated overnight at 4°C while being mixed end over end. Next, the sample was added to 150 µl of Dynabeads Protein G (Life Technologies, Carlsbad, California) washed in immunoprecipitation buffer. Samples were incubated for 2 hr at 4°C while being mixed end over end.

Next, the Dynabeads Protein G and immunoprecipitated DNA were washed. First, the supernatant was removed after applying the sample to a magnet for 3 min. Next, beads were washed three times in 750 µl of immunoprecipitation buffer. Each wash lasted 10 min and during the wash the samples were mixed end over end at 4°C. On each occasion, the supernatant was removed after applying the sample to a magnet for 3 min.

Next, the bisulfite reaction was completed. First, we added 150 µl of 0.3 N NaOH to the samples. The samples were then incubated for 20 min at 37°C. This step should complete the bisulfite reaction. We then neutralized the NaOH by adding 150 µl of 0.3 N HCl and 17.5 µl of 1 M Tris pH 8.0.

Next, we eluted the immunoprecipitated DNA. First, 0.5 µl of 100 U/µl RNase I was added to each sample. Samples were then incubated for 20 min at 50°C. RNAs still bound to DNA were degraded by adding 1 µl of 0.5 M $MgCl_2$ and 3 µl of 5 U/µl RNase H (New England Biolabs, Ipswich, Massachusetts) to each sample. The samples were then incubated for 20 min at 37°C. To ensure that all DNA fragments were eluted, we added 2 µl of 20 µg/µl of Proteinase K to each sample. The samples were then incubated for 1 hr at 50°C. Finally, a magnet was added to each sample for 3 min and the supernatant containing the eluted DNA was transferred to a new microcentrifuge tube.

Next, DNA was extracted from the supernatant using phenol/chloroform. Prior to adding phenol/chloroform, 150 µl of water was added to each sample. Samples were then combined with 500 µl of phenol/chloroform. Samples were mixed thoroughly and then centrifuged at 18400 g for 5 min. 400 µl of the aqueous phase was transferred to a new microcentrifuge tube and then 400 µl of chloroform was added to this aqueous phase. These samples were mixed and then centrifuged at 18400 g for 2 min. The supernatant was transferred to a new microcentrifuge tube. Finally, the DNA in the sample was ethanol precipitated in the presence of 20 µg of glycogen for 48 hr.

## Triptolide treatment of cells

In order to treat cells with triptolide (R and D Systems, Minneapolis, Minnesota), 20 mM triptolide was prepared in dimethyl sulfoxide.

Triptolide was added to the media covering the MCF-7 cells at a concentration of 1 µM. The plates containing the cells were gently swirled to evenly distribute the triptolide. The plates were then incubated for 2 hr at 37°C. Following this triptolide treatment, bisDRIP-seq was performed as described above.

Each of the two triptolide-treatment biological replicates was performed on a separate plate of cells.

## Input bisDRIP-seq protocol

Two input bisDRIP-seq samples were prepared using the bisDRIP-seq protocol until the S9.6 antibody immunoprecipitation step. Instead of adding the digested DNA to S9.6 antibody, 15 µl of 3 N NaOH was added to 0.2 µg of digested DNA in 135 µl of water. The samples were then incubated for 20 min at 37°C. This step should complete the bisulfite reaction. We then neutralized the NaOH by adding 150 µl of 0.3 N HCl and 17.5 µl of 1 M Tris pH 8.0.

Next, we repeated the elution treatment that was applied to the bisDRIP-seq samples. First, 0.5 µl of 100 U/µl RNase I was added to each sample. Samples were then incubated for 20 min at 50°C. RNAs still bound to DNA were degraded by adding 1 µl of 0.5 M $MgCl_2$ and 3 µl of 5 U/µl RNase H (New England Biolabs, Ipswich, Massachusetts) to each sample. The samples were then incubated for 20 min at 37°C. Next, we added 2 µl of 20 µg/µl of Proteinase K to each sample. The samples

were then incubated for 1 hr at 50°C. Finally, the DNA in the sample was extracted using phenol-chloroform. After this point, the bisDRIP-seq protocol was followed as described above.

## RNase H bisDRIP-seq protocol

Both an RNase H-treated sample and matched control sample were prepared using a single plate of MCF-7 cells. These cells were treated as described in the initial steps of the bisDRIP-seq protocol described above. These initial steps included all steps until and including fragment ion of the DNA with restriction enzymes and confirmation of the digest with an agarose gel,

After those initial steps and immediately prior to immunoprecipitating RNA-DNA hybrids with S9.6 antibody, our sample was split into two samples. RNase H was added to one sample to a final concentration of 0.05 U/µl. We now call this sample the 'RNase H-treated sample'. Dithiothreitol (at the same concentration as in the RNase H enzyme solution) was added to the other sample to a final concentration of 10 µM. RNase I was also added to both samples to a final concentration of 0.1 U/µl in order to prevent non-specific interactions between RNAs in solution and the S9.6 antibody (*Phillips et al., 2013*). Samples were then incubated for 2 hr at 37°C. Following this treatment, both samples were incubated with S9.6 antibody and Dynabeads Protein G as described above in the bisDRIP-seq protocol.

Next, we adjusted the immunoprecipitation wash steps to allow for further RNase H treatment. First, three wash steps were performed in immunoprecipitation buffer as described in the standard bisDRIP-seq protocol. Next, we added two additional wash steps using 200 µl of immunoprecipitation buffer supplemented with 3 mM MgCl$_2$. In each of these final two wash steps, RNase H was added to a final concentration of 0.5 U/µl to the 'RNase H' treated sample, while dithiothreitol was added to a final concentration of 50 µM to the matched control sample. Each wash lasted 20 min and during each wash the samples were mixed end over end at 37°C. On each occasion, the supernatant was removed after applying the sample to a magnet for 3 min.

After these wash steps, the samples were treated with NaOH and eluted as described in the bisDRIP-seq protocol above.

## Delayed bisDRIP-seq protocol

The delayed bisDRIP-seq sample was prepared largely following the bisDRIP-seq protocol, with a few modification.

After cells were washed in PBS as described in the bisDRIP-seq protocol, 5 ml of lysis buffer was added. This lysis buffer did not include bisulfite and was instead composed of 1% SDS, 50 mM Tris pH 8, and 10 mM EDTA. Next, 1 ml of 20 µg/µl Proteinase K was added to the sample. Samples were then incubated overnight at 37°C while being rotated end over end.

Next, nucleic acids were extracted using phenol-chloroform and ethanol precipitated as described in the bisDRIP-seq protocol.

At this point, we treated the nucleic acids with bisulfite. First, the nucleic acids were dissolved in 3.3 mM EDTA. Next, 900 µl of ammonium bisulfite solution and 26.8 µl hydroquinone were added to the sample. The sample was then incubated in a shaking incubator for 2.5 hr at 37°C at 1100 rpm, while being protected from light.

Next, the bisulfite was removed from the samples through dialysis. First, samples were added to 2.5 ml of buffer 3.1 lacking BSA (100 mM NaCl, 50 mM Tris-HCl pH 8.0, 10 mM MgCl$_2$) in a 15 ml conical centrifuge tube. Samples were then added to an Amicon Ultra-4 Centrifugal Filter Unit with a 30,000 nominal molecular weight limit (EMD Millipore, Darmstadt, Germany). Next, the samples were centrifuged following the instructions of the centrifugal filter unit manufacturer. After centrifugation, fresh buffer 3.1 lacking BSA was added to the centrifugal filter unit and centrifugation was repeated. This dialysis process was repeated until the sample was nominally dialyzed at least 1000000 fold. At this point, the samples were resuspended in 850 µl of buffer 3.1 and the remaining steps of the bisDRIP-seq protocol were followed as described above.

## DNA library preparation

Prior to constructing DNA sequencing libraries, the eluted DNA was fragmented into approximately 300 bp fragments. DNA from bisDRIP-seq reactions was re-suspended in 150 µl of 1XTE buffer (10 mM Tris-HCl pH 8.0, 1 mM EDTA). Samples were then fragmented using an S2-series Covaris

ultrasonicator (Woburn, Massachusetts). Sonication was performed at 4°C using the following conditions: intensity 5, 10% Duty Intensity Factor, 200 cycles per burst and 140 s total treatment time.

We next used a Pico-Methyl Seq Library Prep Kit (Zymo Research, Irvine, California) to create the DNA sequencing library. This kit is primarily designed for mapping 5-methylcytosine in small quantities of genomic DNA. We omitted the bisulfite-treatment steps. Instead, we specifically followed the library preparation steps as instructed by the manufacturer. These steps amplified our DNA and used random priming to add indexes to our DNA library that are compatible with Illumina's TruSeq technology.

In preparation for sequencing, the quality of each DNA library was assessed using a 2100 Bioanalyzer (Agilent Technologies, Santa Clara, California) by the Weill Cornell Epigenomics Core (New York City, New York). The concentration of DNA was measured using a Quant-iT dsDNA Assay Kit and a SpectraMax M2 Microplate Reader following the instructions provided in the Quant-iT dsDNA Assay kit.

DNA libraries were sequenced by the Weill Cornell Epigenomics Core using a HiSeq 2500 System (Illumina, San Diego, California). Either five or six samples were loaded per lane. DNA libraries were sequenced following the manufacturer's instructions for single-index 100 bp paired-end read clustering.

## Initial data processing

Sequencing data was first processed using CASAVA 1.8.2 (Illumina, San Diego, California) to obtain the nucleotide sequence of reads in FASTQ format.

We next aligned reads to the GRCh38 reference genome. Reads were aligned to the genome using the Bismark software library version 0.14.3 Bismark tool (*Krueger and Andrews, 2011*). The Bismark software library is specifically designed to align bisulfite-modified reads. In this analysis, the Bismark library used the Bowtie2 alignment tool version 2.2.5 (*Langmead and Salzberg, 2012*) to perform the actual alignments. We also selected the Bismark option for post-bisulfite adapter tagging.

Read alignment was done in three phases. If we were unable to map a read in one phase, then we attempted to map the read in the following phase.

In the first alignment phase, full paired-end sequencing reads were aligned to the genome. First, reads were trimmed to remove both low confidence sequences and the very ends of reads. This was done using Flexbar version 2.5 (*Dodt et al., 2012*) with the following settings: quality format i1.8, phred minimum of 30, 10 nt removed from 5' end of reads. Next, the Bismark tool was used to align reads to the genome. The settings employed for the alignment did not allow for mismatched nucleotides and restricted the gap between paired-end reads to no more than 1000 bp.

Next, the Bismark alignment tool was repeated on all reads that failed to map in the first alignment phase. In this second phase, the two ends of paired-end reads were treated as single-end reads. The settings employed for alignment did not allow mismatched nucleotides.

In the final alignment phase, we attempted to align all reads that were not aligned to the genome in the first two phases after additional preprocessing. In this phase, we continued to treat both ends of paired-end reads as single-end reads. However, we began this phase by using Flexbar version 2.5 to remove all nucleotides 55 nt or further 3' of the start of the read. Reads were then aligned with the Bismark tool using settings that did not allow for mismatched nucleotides.

Having aligned reads to the genome, we next removed putative read duplicates. To remove read duplicates, we ran the Bismark deduplicate_bismark tool.

Finally, we determined if cytosines within aligned reads had been converted by bisulfite. To determine whether individual cytosines had been converted to uracils, we used the Bismark bismark_methylation_extractor tool. In the case of paired-end alignments, we used the –no_overlap setting. With single-end alignments, we used the –ignore_3prime 10 setting. This setting removed the final ten nucleotides of each alignment. These nucleotides were removed from aligned reads because we observed bias in the fraction of cytosines converted in this region of reads.

## Calculation of raw bisDRIP-seq read scores

Next, we developed a method to score reads based on the likelihood that a given read was single-stranded in our samples. This measure was intended to filter out conversions that occur due to

spontaneous breathing of DNA. Additionally, we wanted to ensure that single-stranded reads containing large numbers of cytosines were not given greater value than single-stranded reads containing small numbers of cytosines. This method was applied to on each sample separately.

For the purposes of calculating bisDRIP-seq scores, we define the 'original number of cytosines' as the sum of converted cytosines and unconverted cytosines.

The goal of this method was to estimate the fraction of reads that had a set number of conversions due to chance alone. The inverse fraction was then defined as the bisDRIP-seq score for all reads with that number of conversions. Therefore, all of the reads from the same sample, with the same original number of cytosines, and with the same number of converted cytosines were given the same bisDRIP-seq score.

First, we made a rough estimate of the background conversion rate for each sample. Initially, we calculated the total fraction of cytosines that were converted in our aligned reads. Next, we repeated this calculation after removing reads that had greater than 2.5 fold more conversions than the average across all reads. This value was used as an estimate of the background conversion rate.

Next, we calculated the probability of observing a specified number of converted cytosines in a given read by chance. This probability was calculated using the binomial distribution. In this binomial distribution calculation, the size of the sample was the original number of cytosines in the read and the probability of a conversion was the estimated background conversion rate.

Next, we calculated the number of reads expected to have a given original number of cytosines and a given number of converted cytosines. First, we calculated the total number of aligned reads with a given original number of cytosines. Next, we multiplied that number with the probability of observing the specified number of converted cytosines in a read with that original number of cytosines by chance.

Next, a bisDRIP-seq score was calculated for each read. First, reads were binned together based on the original number of cytosines in the read and the number of converted cytosines in the read. Next, bins were grouped together if they had the same original number of cytosines. If a group of reads with a given original number of cytosines per read consisted of more than a thousand reads in total and the expected number of reads for each bin was above five, then we calculated the bisDRIP-seq for each bin as:

$$if\ O > E,\ bisDRIPseq\ score = 1 - \frac{E}{O}\quad \text{E} = \text{expected number of reads}$$
$$if\ O \leqslant E,\ bisDRIPseq\ score = 0\quad \text{O} = \text{observed number of reads}$$

This value was calculated first for reads with one original number of cytosines and then was calculated for reads with progressively higher numbers of original cytosines. If the expected number of reads for a bin was below five, then the bisDRIP-seq score was calculated for all reads with x number of converted cytosines using the equation:

$$if \sum_{c=x}^{n} E_C > 5: \quad \begin{array}{l} if \sum_{c=x}^{n} O_c > \sum_{c=x}^{n} E_c,\ bisDRIPseq\ score_x = 1 - \frac{\sum_{c=x}^{n} E_c}{\sum_{c=x}^{n} O_c} \\ if \sum_{c=x}^{n} O_c \leqslant \sum_{c=x}^{n} E_c,\ bisDRIPseq\ score_x = 0 \end{array}$$
$$if \sum_{c=x}^{n} E_c \leqslant 5: \quad bisDRIPseq\ score_x = bisDRIPseq\ score_{x-1}$$
$$c = number\ of\ cytosine\ conversions$$
$$n = original\ number\ of\ cytosines$$
$$E_c = expected\ number\ of\ reads\ with\ c\ cyosine\ conversions$$
$$O_c = observed\ number\ of\ reads\ with\ c\ cyosine\ conversions$$

In this equation, the original number of cytosines (n) is held constant.

If there were fewer than a thousand reads in a group, reads were scored differently. bisDRIP-seq scores with a specific number of conversions were given the same score as reads with the same number of conversions, but one fewer original number of cytosines. Given the total number of aligned reads per sample, the number of reads assigned scores on this basis was relatively small.

Source code for calculation of bisDRIP-seq scores from read sequence is in *Source code 1* and processingbisDRIPseqreads.py, which has been deposited in https://github.com/champben2002/bisDRIPseq/

## Normalization of bisDRIP-seq read scores

bisDRIP-seq scores were normalized to ensure that the sum of the bisDRIP-seq scores were the same across samples after normalization. This normalization procedure assumes that there are no global differences in the amount of single-stranded structure between samples. First, the sum of all read bisDRIP-seq scores was calculated for each sample. Next, the normalized bisDRIP-seq score for each read was calculated for each read in a given sample using the formula:

$$normalized\ bisDRIPseq\ score\ for\ read\ Y = \frac{\mathrm{bisDRIPseq\ score\ for\ read\ Y} \times 1000000}{\sum \mathrm{all\ read\ bisDRIPseq\ scores}}$$

This normalization procedure was repeated for each sample separately.

All bisDRIP-seq scores in the manuscript used this standard normalization procedure, unless otherwise noted.

Promoter-associated R-loops have previously been associated with promoter regions enriched in guanines on the non-template strand (*Ginno et al., 2012*). We did not make conclusions related to this finding since bisDRIP-seq scores depend on the presence of cytosines in single-stranded regions of reads. However, we did consider whether our conclusions might be affected by the cytosine content of reads.

To test for whether our conclusions were affected by the cytosine content of reads, we applied a 'cytosine normalization' procedure to the raw bisDRIP-seq scores. First, reads within a sample were grouped based on the original number of cytosines in each read. Next, we normalized the reads within each group so that the average bisDRIP-seq score for each group would be identical. This normalization was done by multiplying the bisDRIP-seq score of each read within a group by a group-specific variable. Once this cytosine normalization was complete, the standard normalization procedure was followed as described above.

We repeated our analysis using these cytosine-content normalized scores (*Figure 5—figure supplement 3*). Our conclusions were not changed by this cytosine-content normalization.

Source code for bisDRIP-seq normalization is included in processingbisDRIPseqreads.py, which has been deposited in *Source code 1* and https://github.com/champben2002/bisDRIPseq/

## Display of bisDRIP-seq scores and other data on a genome browser

All genomic maps were generated using the Integrative Genomics Viewer version 2.3.59 (86) with the GRCh38 human genome (*Robinson et al., 2011*; *Thorvaldsdóttir et al., 2013*). The automatically loaded refseq gene models (*O'Leary et al., 2016*) were included in each map, with the following exceptions: For *NEAT1*, we used the gene model from GENCODE version 24 (*Harrow et al., 2012*) of the long isoform of *NEAT1*, ENST00000501122.2, since it appears more consistent with the RNA-seq data. For *HIST1H2BK*, we used the gene model from GENCODE version 24 (*Harrow et al., 2012*) of the intronless isoform of *HIST1H2BK*, ENST00000356950.1.

bisDRIP-seq scores were calculated for each nucleotide in the genome. The score of a given nucleotide in the genome was calculated as the sum of the bisDRIP-seq scores of reads that aligned to that nucleotide. Notably, this score was strand specific. The bisDRIP-seq score of reads was associated with a given nucleotide only if that nucleotide was on the same strand as the cytosines converted in the read. These bisDRIP-seq scores were then plotted for specific genomic loci.

Local sequence composition was plotted for sites in the *HIST1H1E and HIST1H2BG* genes. At each site, the mean number of adenines, cytosines, guanines and thymines in the surrounding region were plotted. The surrounding region included the nucleotide at the site, the twenty-five nucleotides 5' of the site and twenty-five nucleotides 3' of the site on the template strand.

## Calculation of bisDRIP-seq scores for genomic regions

bisDRIP-seq scores were calculated for genomic regions using the following procedure:

First, we summed the bisDRIP-seq scores of reads that completely aligned to the specified region.

Next, reads partially in the specified region were partially added to the specified region's score. The fraction of the read's score added to the score of the specified region's score was determined by the fraction of the read's length that aligned to the region. Thus the fraction of the read's length contained within the region was multiplied by the read's score. The product of this calculation was

added to the specified region's score. This was repeated for each read that partially aligned in the specified region. This procedure provided the region's final bisDRIP-seq score.

In some of our analysis, bisDRIP-seq scores were calculated for only one strand within a specified region. In this case, the bisDRIP-seq scores from individual reads was only added to the specified region's bisDRIP-seq score if the read had the correct strand orientation. A read had the correct strand orientation if any converted cytosines in the read mapped to the correct strand.

Source code for calculating region scores is incorporated into regionbisDRIPseqscores.py, which has been deposited in *Source code 2* and https://github.com/champben2002/bisDRIPseq/.

## Monte Carlo simulations

Monte Carlo simulations were used to determine the expected distribution of bisDRIP-seq scores if they distributed randomly across the genome. Two simulations were performed.

First, a simulation was performed in which reads were mapped to random regions across the genome. This simulates the possibility that both the RNA-DNA hybrid immunoprecipitation and the bisulfite mapping in bisDRIP-seq are non-specific. First, we split the genome into 1 kb regions. Next, we determined if any bisDRIP-seq reads aligned to a given 1 kb region. If no reads aligned to a given region, then that region was removed from further simulation steps. Reads were then associated with a random 1 kb region that was in the same chromosome to which they originally mapped. The region that each read was associated with was determined using the R 'random' package. This was repeated for each of our thirteen samples.

Second, a simulation was performed in which bisDRIP-seq scores were randomly shuffled between reads. This simulates the possibility that the bisDRIP-seq score associated with each read was stochastic. For each sample, we randomly shuffled all of the scores associated with reads aligned to a given chromosome. The random shuffle of read scores was performed using the R 'random' package. This simulation was repeated for each chromosome and for each of the thirteen bisDRIP-seq samples.

Next, bisDRIP-seq scores were calculated for each 1 kb region using either real bisDRIP-seq scores or using the products of one of the two simulations. This analysis used the same 1 kb regions discussed in the first Monte Carlo simulation.

Source code for Monte Carlo simulations are included in Monte_Carlo_random_assign_reads_to_regions.py and Monte_Carlo_for_shuffling_bisDRIPseq_scores.py, which have been deposited in *Source code 3* and *Source code 4* (respectively), as well as https://github.com/champben2002/bisDRIPseq/

## Creating a reference set of transcription start sites

GENCODE's annotated list of transcription start sites (*Harrow et al., 2012*) was used to generate a 'reference GENCODE transcription start site list.' This set of transcription start sites was then used to define promoter regions in the genome and as reference points for metaplots.

First, the GENCODE comprehensive gene annotation release 25 was downloaded from GENCODE's website.

Next, transcription start sites were removed from the transcription start site list if they were within a thousand base pairs of a higher priority transcription start site. The priority of transcription start sites was determined as follows:

1. The highest priority transcription start sites were transcription start sites within 25 bp of a site identified by CAGE-seq. Transcription start sites near sites that had higher CAGE-seq signal were given a higher priority than transcription start sites near sites with lower CAGE-seq signal.
2. Priority was next determined by the confidence annotation scores associated with each transcription start site in the GENCODE annotations.
3. Finally, priority was established using a random number generated using the Python 'random' package.

After removing lower priority transcription start sites, we had a final reference GENCODE transcription start site list. Promoter regions were then defined as the region within a thousand base pairs of each transcription start site in our reference GENCODE transcription start site list. See Source_data_file_1.xls for all transcription start sites in this final list

In later analysis, inactive promoter regions were removed. Unless noted otherwise, inactive promoter regions were defined as promoter regions with sense-strand promoter activity in the bottom eighty percentile between the transcription start site and + 1000 bp.

## Calculating promoter activity

In order to calculate the promoter activity of each promoter, we used publicly available GRO-seq data obtained from MCF-7 cells by *Hah et al., 2013*. In particular, data was combined from files GSM1067410, GSM1067411, GSM1067412, GSM1067413, GSM1067414 and GSM1067415. The MCF-7 cells used in these samples were treated in a similar manner to the MCF-7 cells used in our bisDRIP-seq protocols.

Unless noted otherwise, promoter activity was defined by the mean number of sense-strand GRO-seq reads measured in the region between the transcription start site and + 1000 bp. The method for calculating the number of reads in this region of each promoter was similar to method used to calculate bisDRIP-seq scores for regions.

For each promoter region, we first summed the number of GRO-seq reads that completely aligned to the region between the transcription start site and + 1000 bp.

Next, reads partially in the specified region were partially added to the specified region's score. The fraction of the read's length contained within the region between the transcription start site and + 1000 bp was added to the promoter region's score. This was repeated for each read that partially aligned to the promoter region. This procedure provided the promoter region's final promoter activity.

A similar procedure was followed to determine the antisense promoter activity and the promoter activity between the transcription start site and + 250 bp.

## Creating a reference set of exon-intron junctions and calculating first-exon lengths

Exon-intron junctions were extracted from the GENCODE comprehensive gene annotation release 25 (*Harrow et al., 2012*). Exon-intron and intron-exon junctions were separated based on whether they were the first exon-intron, first intron-exon or second exon-intron junction in a given annotated transcript. If two promoter regions shared the same exon-intron junction, then one was removed from the list.

Next, all exon-intron junctions were removed that were associated with inactive promoter regions. Inactive promoter regions did not contain positive R-loop signal in *Figure 3F*, suggesting that removing these promoter regions would lead to a greater signal-to-noise ratio. The activity of promoter regions was defined here based on the number of sense-strand GRO-seq reads that aligned to the promoter region between the transcription start site and + 1000 bp. Promoters in the bottom eighty percentile in terms of promoter activity were considered inactive promoters and removed from exon-intron junction analysis.

Exon-intron junctions associated with the intronless genes, *MALAT1* and *NEAT1*, were also removed.

This procedure provided a final reference set of exon-intron junctions. See *Figure 4—source data 1* for this final reference list.

First-exon length was calculated for each first exon-intron junction in the final reference set of first exon-intron junctions. The first-exon length was calculated as the distance between the transcription start site and the first exon-intron junction. This list of first-exon lengths was used to calculate the median length of all first exons for active promoter regions.

## Additional external sources of data

In order to compare some external dataset to our data, it was necessary to convert the coordinates of sites or regions to the GRCh38 genome from an earlier version of the reference human genome. In all cases, this was accomplished using the UCSC utility liftOver (*Speir et al., 2016*).

Cap analysis gene expression sequencing (CAGE-seq) data came from the publicly-available ENCODE MCF-7 dataset ENCFF207DXM (*ENCODE Project Consortium, 2012*).

In order to compare bisDRIP-seq datasets with DRIPc-seq datasets, we used the DRIPc-seq data of *Sanz et al., 2016*. In particular, we compared our data against the data deposited in the Gene Expression Omnibus GSM1720613 and GSM1720614 wig files.

In order to compare bisDRIP-seq datasets with DRIP-seq datasets, we used the DRIP-seq data of *Sanz et al., 2016*. In particular, we compared our data against the data deposited in the Gene Expression Omnibus GSM1720615 and GSM1720616 wig files.

Total cell RNA-seq data displayed for individual genes came from ENCODE files ENCFF426WXY and ENCFF866OVQ (*ENCODE Project Consortium, 2012*). The RNA-seq experiments used to generate that data were rRNA depleted and used MCF-7 cells.

Nucleus RNA-seq levels of active genes in *Figure 5—figure supplement 1A* are the mean transcripts per kilobase million (TPM) values as measured in ENCODE files ENCFF063BLU and ENCFF285GOS (*ENCODE Project Consortium, 2012*). These RNA-seq experiments sequenced mRNA depleted in rRNA from the nuclei of MCF-7 cells.

RNA Polymerase II levels were calculated using the ENCODE files ENCFF496YAE and ENCFF881YOO (*ENCODE Project Consortium, 2012*). These datasets comes from chromatin immunoprecipitations of POLR2A in MCF-7 cells. The number of reads between the transcription start site and + 250 bp of each gene was calculated using the same methodology described above for GRO-seq reads.

## Promoter region enrichment tests

For DRIP-seq and bisDRIP-seq promoter region enrichment tests, only promoter regions in genes larger than 2 kb were considered. For each promoter region, a matched region was selected from the same gene. This matched region was centered on an exonic site more than 2 kb from the transcription start site. Promoter enrichment was then tested using the non-parametric Wilcoxon signed-rank test described below.

## Ranking promoter regions by how strongly they associate with transcription-dependent R-loops

Promoter region ranking was only performed on active promoter regions. Promoter activity was defined here based on the number of sense-strand GRO-seq reads that aligned to the promoter region between the transcription start site and +250 bp. Active promoter regions were promoter regions in the top twenty percentile of promoter regions in terms of promoter activity. This filter was intended to filter out inactive promoter regions, which appeared to only contribute noise to our analysis.

Next, promoter regions were filtered on the basis of sense-strand bisDRIP-seq scores. The non-template strand bisDRIP-seq score had to be positive in the region between the transcription start site and + 250 bp in all control-treated biological replicates. Otherwise, the promoter region was removed from out list.

The remaining promoter regions were scored based on the value:

2 x (log2(control-treated non-template bisDRIP-seq score + 1) - log2(triptolide-treated non-template bisDRIP-seq score + 1)) - log2(control-treated template bisDRIP-seq score + 1))

## Generating graphical plots

Loess smoothed plots were generated using the r fANCOVA package's loess.as algorithm with parameters 'criterion = aicc' and 'family = gaussian'.

Scatterplots were generated with the ggplot2 package geom_points() algorithm, using the lm option to plot a linear regression line of the displayed points.

Violin plots and jitter plots were created using the ggplot2 package. Jitter plots were created using geom_jitter() with height = 0, width = 0.725. Violin plots were created using geom_violin() with alpha = 0.5.

bisDRIP-seq score metaplots were created in relation to reference points. These reference points were either transcription start sites or exon-intron junctions, depending on the analysis. The final metaplot associated a bisDRIP-seq score to each 'location' relative to a given reference point. Location refers to the distance from the reference point and whether the site is upstream or downstream of the reference site. For a given read, we determined the location of each nucleotide in the read

relative to the proximal reference point. Next, the location of each nucleotide was associated with the bisDRIP-seq score of the read. After repeating this process for all reads, all of the scores associated with a given location were summed unless otherwise noted. This sum was then used as the bisDRIP-seq score for that location in the metaplot. Finally, each location was plotted against the bisDRIP-seq score associated with that location.

Source code for metaplot analysis is incorporated into bisDRIPseqmetaplotanalysis.py, which has been deposited in *Source code 5* and https://github.com/champben2002/bisDRIPseq/.

The conversion asymmetry metaplot was generated in relation to transcription start sites. The total number of conversions was summed at each site relative to the transcription start site across all active promoter regions. This was repeated for each sample, strand and sample type. Next, we calculated the mean number of conversions across samples for each position relative to the transcription start site. Next, we subtracted the mean number of conversions for each position relative to the transcription start site in triptolide-treated samples from the number of conversions observed in control-treated samples. Finally, we plotted the log ratio of the tripolide-corrected number of conversions on the non-template strand to the tripolide-corrected number of conversions on the template strand.

To create a metaplot of the percentage of cytosines converted on the template strand, we calculated the number of cytosines and converted cytosines that aligned to a given site relative to the transcription start site. Next, the number of converted cytosines was divided by the total number of original cytosines that aligned to that site.

DRIPc-seq metaplots were generated using essentially the same process as was used for bisDRIP-seq score. Instead of using the scores of reads, we used the read-count associated with regions in the wig file.

In all plots of R-loop signal, R-loop signal = (control-treated sample, non-template bisDRIP-seq score - triptolide-treated sample, non-template bisDRIP-seq score) - (control-treated sample, template bisDRIP-seq score - triptolide-treated sample, template bisDRIP-seq score).

In all plots where a log2 transformation was applied to a dataset, each value in the dataset was added to one prior to the log2 transformation.

## Sampling *MALAT1* reads from the *MALAT1* R-loop forming region

To sample the reads from the *MALAT1* R-loop forming region, we selected all reads that: (1) aligned to the non-template strand of *MALAT1,* (2) start and end between the transcription start site and + 1600 bp, and (3) are larger than 75 bp long. Next, we randomly selected twenty-five reads. Reads were then plotted using ggplot2's geom_point function.

## Statistical tests

Since we are not confident that our data follows a Gaussian distribution, we typically used non-parametric tests of significance. In particular, we applied the Wilcoxon signed-rank test whether various datasets were significantly different. Wilcoxon signed-rank tests were performed using the two-sided R wilcox.test() algorithm using default parameters with the exception of the 'paired' setting which was set to 'FALSE'.

In the case of comparisons between metaplots, a conservative multiple-hypothesis correction was performed on the p-values derived from Wilcoxon signed-rank tests. This multiple-hypothesis correction involved multiplying the p-value of an individual nucleotide position by 2001, since the metaplots involved examining 2001 nucleotide positions simultaneously.

The significance of correlations between datasets was examined using the Spearman's rank-correlation test with asymptotic t approximation. Spearman's rank-correlation tests were used since we are not confident that our data follows a Gaussian distribution and therefore wished to apply a non-parametric test of correlation. Spearman's rank-correlation tests were performed using the R cor.test (x, y, method = 'spearman') algorithm.

## Data accession information

For each sample, we deposited the sequence reads, the conversion frequency at each cytosine nucleotide and a bedGraph of bisDRIP-seq scores in GEO series GSE98886: https://www.ncbi.nlm.nih.gov/geo/query/acc.cgi?acc=GSE98886

Private token for reviewers: azyfacgczpertip

## Acknowledgements

We thank C Kam and members of the Jaffrey laboratory for comments and suggestions, the members of the Weill Cornell Epigenomics Core for their assistance in high-throughput sequencing, and the Mittal lab (Weill-Cornell) for their assistance with DNA sonication. This work was supported by a Tri-Institutional Starr Stem Cell Scholars Fellowship (JD) and NIH grant R01 CA186702 (SRJ).

## Additional information

### Funding

| Funder | Grant reference number | Author |
|---|---|---|
| Starr Foundation | WC2015-011 | Jason G Dumelie |
| National Institutes of Health | R01 CA186702 | Samie R Jaffrey |

The funders had no role in study design, data collection and interpretation, or the decision to submit the work for publication.

### Author contributions

Jason G Dumelie, Conceptualization, Data curation, Software, Formal analysis, Funding acquisition, Investigation, Methodology, Writing—original draft; Samie R Jaffrey, Conceptualization, Resources, Supervision, Funding acquisition, Investigation, Methodology, Writing—original draft, Writing—review and editing

### Author ORCIDs

Jason G Dumelie  http://orcid.org/0000-0002-6926-4843
Samie R Jaffrey  http://orcid.org/0000-0003-3615-6958

### Decision letter and Author response

Decision letter https://doi.org/10.7554/eLife.28306.057
Author response https://doi.org/10.7554/eLife.28306.058

## Additional files

### Supplementary files

• Source code 1. Calculating bisDRIP-seq scores from bisDRIP-seq reads.
DOI: https://doi.org/10.7554/eLife.28306.031

• Source code 2. Calculating region bisDRIP-seq scores.
DOI: https://doi.org/10.7554/eLife.28306.032

• Source code 3. Stochastic read alignment Monte Carlo simulations.
DOI: https://doi.org/10.7554/eLife.28306.033

• Source code 4. Stochastic read score switching Monte Carlo simulations.
DOI: https://doi.org/10.7554/eLife.28306.034

• Source code 5. Generating metaplots around specific sites.
DOI: https://doi.org/10.7554/eLife.28306.035

• Transparent reporting form
DOI: https://doi.org/10.7554/eLife.28306.036

### Major datasets

The following dataset was generated:

| Author(s) | Year | Dataset title | Dataset URL | Database, license, and accessibility information |
|---|---|---|---|---|
| Dumelie JG, Jaffrey SR | 2017 | Defining the location of promoter-associated R-loops at near-nucleotide resolution using bisDRIP-seq | https://www.ncbi.nlm.nih.gov/geo/query/acc.cgi?acc=GSE98886 | Publicly available at the NCBI Gene Expression Omnibus (accession no. GSE98886) |

The following previously published datasets were used:

| Author(s) | Year | Dataset title | Dataset URL | Database, license, and accessibility information |
|---|---|---|---|---|
| Sanz LA, Lim Y, Hartono SR, Raj-purkar AR, Chedin F | 2016 | Human and mouse DRIP-seq and DRIPc-seq | https://www.ncbi.nlm.nih.gov/geo/query/acc.cgi?acc=GSE70189 | Publicly available at the NCBI GenBank (accession no. GSE70189) |
| ENCODE (Carninci Lab) | 2012 | CAGE on human MCF-7 nucleus long polyadenylated RNA | https://www.encodeproject.org/experiments/ENCSR000CKV/ | Publicly available at ENCODE (accession no: ENCFF207DXM) |
| ENCODE (Graveley Lab) | 2016 | RNA Evaluation MCF7 Long Total from Graveley | https://www.encodeproject.org/experiments/ENCSR667JTA/ | Publicly available at ENCODE (accession no: ENCFF426WXY) |
| ENCODE (Graveley Lab) | 2016 | RNA Evaluation MCF7 Long Total from Graveley | https://www.encodeproject.org/experiments/ENCSR667JTA/ | Publicly available at ENCODE (accession no: ENCFF866OVQ) |
| Hah N, Murakami S, Nagari A, Danko CG, Kraus WL | 2013 | Enhancer Transcripts Mark Active Estrogen Receptor Binding Sites | https://www.ncbi.nlm.nih.gov/geo/query/acc.cgi?acc=GSE43835 | Publicly available at the NCBI Gene Expression Omnibus (accession no. GSE43835) |
| ENCODE (Yeo Lab) | 2015 | eCLIP experiment on HepG2 against SRSF1 | https://www.encodeproject.org/experiments/ENCSR989VIY/ | Publicly available at ENCODE (accession no: ENCFF327NVE) |
| ENCODE (Yeo Lab) | 2015 | eCLIP experiment on K562 against SRSF1 | https://www.encodeproject.org/experiments/ENCSR432XUP/ | Publicly available at ENCODE (accession no: ENCFF137IAG) |
| ENCODE (Iyer Lab) | 2011 | POLR2A ChIP-seq on human MCF-7 | https://www.encodeproject.org/experiments/ENCSR000DMT/ | Publicly available at ENCODE (accession no: ENCSR000DMT) |
| ENCODE (Gingeras Lab) | 2012 | RNA-seq (polyA mRNA RNA-seq) on Homo sapiens MCF-7 nuclear fraction | https://www.encodeproject.org/experiments/ENCSR000CTO/ | Publicly available at ENCODE (accession no: ENCSR000CTO) |

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
