## [Decision Letter]

Thank you for submitting your article "Defining the location of promoter-associated R-loops at near-nucleotide resolution using bisDRIP-seq" for consideration by *eLife*. Your article has been reviewed by three peer reviewers, and the evaluation has been overseen by a Reviewing Editor and Kevin Struhl as the Senior Editor. The reviewers have opted to remain anonymous.

The reviewers have discussed the reviews with one another and the Reviewing Editor has drafted this decision to help you prepare a revised submission.

As stated by the reviewers the new method to map the position of R-loops at higher resolution than existing techniques is important to the field. The approach entails converting cytosines to uracils in the displaced DNA strand with bisulfite, while the cytosines in the opposite strand are protected by RNA. However all reviewers felt that it is crucial to include an RnaseH control. In addition since this is a methods paper it is necessary to discuss the differences and advantages of the reported method to those that have been previously developed and how the method and/or results represent a significant advance over the literature. We look forward to see an improved manuscript.

Reviewer #1:

R-loops are three-stranded structures consisting of a DNA-RNA hybrid and a displaced single-stranded DNA. R-loops are involved in multiple cellular processes, including mitochondria DNA replication, immunoglobulin class switch recombination, and transcription regulation. However, when dysregulated, R-loops could challenge genome integrity by introducing mutations, single-strand breaks, stalled replication forks and double-strand breaks. Precise genome-wide mapping of R-loops may provide insights into their functions and dynamics.

In this manuscript, the authors developed a new method to map R-loops at near-nucleotide resolution. Their new method, named bisDRIP-seq, takes advantage of the unique three-stranded structures of R-loops. After fragmentation, RNA-DNA hybrids are pulled down using antibody S9.6, and cytosines in the displaced single-stranded DNA are converted to uracils by bisulfite. The cytosine-uracil-conversions can be mapped to the genome at near-nucleotide resolution. While this reviewer finds the approach novel, it does not appear to provide significantly higher resolution over the existing DRIPc-seq method developed recently by Chedin and coworkers (PMID: 27373332). Major concerns are listed below. In particular, baseline cytosine-uracil-conversions without S9.6 pull-down should be performed for the purpose of background elimination. Furthermore, as stated below, RNase H treatment prior to immunoprecipitation should be performed to validate the specificity of the bisDRIP-seq signals.

Major concerns:

1) It is crucial to confirm signals from bisDRIP-seq are indeed R-loop dependent. Simple bisDRIP-seq and DRIP-seq correlation is not sufficient. BisDRIP-seq with in vitro RNase H treatment should be performed.

2) In bisDRIP-seq, signals not only come from R-loops, but also from promoter single-stranded structures. Therefore it is important to include non-denatured cytosine-uracil-conversions (without S9.6 pull-down) as bisDRIP-seq input.

3) Template-strand bisDRIP-seq scores were subtracted from non-template-strand bisDRIP-seq scores to artificially recover R-loop signals. However, no assay was used to prove the validity of such a method.

4) Since previously reported DRIPc-seq already provides strand-specific, near base-pair resolution (PMID: 27373332), basic comparisons should be performed to test whether bisDRIP-seq outperforms DRIPc-seq in a significantly manner.

Reviewer #2:

The authors describe a new method to map the position of R-loops at higher resolution than existing techniques. The approach entails converting cytosines to uracils in the displaced DNA strand with bisulfite, while the cytosines in the opposite strand are protected by RNA. To minimize the amount of sequencing they focus on regions immunoprecipitated by the S9.6 antibody specific for RNA-DNA hybrids. After bioinformatic processing the authors obtain a genome-wide "bisDRIP-score" which measures the frequency of single-stranded DNA present at a given locus. Asymmetric single-stranded regions are considered R-loops. R-loops are correlated with transcription and enriched in the promoter region of active genes. The high resolution of bisDRIP-seq allowed the authors to identify the TSS as a 5' boundary and the first exon-intron junction as a 3' boundary. Finally, they report that histone genes and lncRNAs MALAT1 and NEAT1 have a particularly strong R-loop signature.

R-loops are believed to play important roles in gene regulation and the manuscript by Dumelie et al. describes a creative strategy to better map their genomic localization. The results are well presented and the methods are described clearly and in much detail. Overall I support publication in *eLife*. There are some major issues that need to be addressed, mostly ensuring that the authors are truly measuring R-loops and that no biases are skewing their analyses.

– BisDRIP score: I find the explanation of how this score was calculated rather convoluted. The choice of using the sum of the bisDRIP-score for cumulative anlayses seems questionable. This makes it impossible to compare axes when different number of regions are analyzed. For example 3C vs. 3E. Why not using the average?

– Validations: because bisDRIP-seq is a new genome-wide technique a high level of confidence in its measurements is required before it can be employed for new discoveries. 1) The authors show significant correlation genome-wide between DRIP-seq (the established technique) and their new bisDRIP-seq, but the correlation is obviously not perfect. Can the authors identify examples in which their new technique and the old technique differ and independently verify that the new mapping is more accurate? 2) To validate DRIP-seq typically RNaseH is used to remove RNA-DNA hybrids and prove the specificity of the signal. I believe this would be a more convincing control than triptolide also for bisDRIP-seq.

– Boundaries: the authors state that the TSS and the first exon-intron junction form 5' and 3' boundaries for R-loop formation, respectively, yet in several figures (e.g. Figure 3, Figure 4) I see signal above background upstream and downstream of these boundaries. Is this because bisDRIP-seq is not exactly single-base resolution, because of smoothing artifacts, or because these are not hard boundaries? Also, it was reported that restriction enzyme digestion can lead to overrepresentation of ORFs and first exon bias (László Halász et al., Genome Res 2017). It is important that the authors exclude this possibility in their own analyses.

– Methylcytosines: it seems that the presence of methylated cytosines would make it difficult to detect R-loops in heavily methylated areas of the genome. Is this a problem? It should be discussed.

– Genes with strong R-loop signals: the authors conclude that not having introns is a reason why some of the genes make in the top 25 list shown in Table 1 but alternative possibilities should be considered: 1) MALAT1, NEAT1 and histone RNAs are not terminated canonically; has that anything to do with R-loops? 2) What is the relationship between steady state RNA levels and formation of R-loops? GRO-seq measures transcription levels, and a weak correlation is visible in Figure 5, but what about total RNA levels? Ribominus RNA-seq to measure steady-state levels of nuclear RNAs might be more suitable for this.

Reviewer #3:

In this manuscript, Dumelie et al. develop a new method, Bis-DRIP Seq, to identify the genomic localization of R loops at near-nucleotide resolution. Given the growing interest in understanding the function of R loops, this new method will benefit research in this area. Using this technique authors show that R loops are associated with the promoter regions of genes with or without introns. However, in genes with introns, 1st exon-intron junction defines the boundary of the R loop. We suggest a few minor experiments before publication of this manuscript.

Specific comments:

1) The authors highlight that BisDRIP-Seq provides more information about R loop boundaries as compared to previous DRIP-Seq datasets. This new method is an extension of the previously established DRIP-Seq and in theory it is understandable why it is an improvement. But since this is a 'methodology paper', the authors must provide comparative evidence that it is better. A figure showing how their dataset compares with a DRIP-Seq dataset will be beneficial.

2) The authors show that the 1st exon-intron junction serves as the 3' R-loop boundary (Figure 4) which they suggest might be due RNA splicing (Figure 7). To extend the biology a little more, the authors should look at a few genes that are known to have alternative splice forms to test if R loop boundaries show differences in genes that are known to have alternative splicing versus those that do not.

---

## [Author Response]

Reviewer #1:R-loops are three-stranded structures consisting of a DNA-RNA hybrid and a displaced single-stranded DNA. R-loops are involved in multiple cellular processes, including mitochondria DNA replication, immunoglobulin class switch recombination, and transcription regulation. However, when dysregulated, R-loops could challenge genome integrity by introducing mutations, single-strand breaks, stalled replication forks and double-strand breaks. Precise genome-wide mapping of R-loops may provide insights into their functions and dynamics.In this manuscript, the authors developed a new method to map R-loops at near-nucleotide resolution. Their new method, named bisDRIP-seq, takes advantage of the unique three-stranded structures of R-loops. After fragmentation, RNA-DNA hybrids are pulled down using antibody S9.6, and cytosines in the displaced single-stranded DNA are converted to uracils by bisulfite. The cytosine-uracil-conversions can be mapped to the genome at near-nucleotide resolution. While this reviewer finds the approach novel, it does not appear to provide significantly higher resolution over the existing DRIPc-seq method developed recently by Chedin and coworkers (PMID: 27373332). Major concerns are listed below. In particular, baseline cytosine-uracil-conversions without S9.6 pull-down should be performed for the purpose of background elimination. Furthermore, as stated below, RNase H treatment prior to immunoprecipitation should be performed to validate the specificity of the bisDRIP-seq signals.

The reviewer points out that our method to map R-loops at near-nucleotide resolution is novel. Still, the reviewer points out that the manuscript would be more compelling if we clearly showed how our method improves upon pre-existing DRIP-seq based techniques. Finally, the reviewer suggests several important experiments that could help us improve readers' confidence in the bisDRIP-seq method.

Major concerns:1) It is crucial to confirm signals from bisDRIP-seq are indeed R-loop dependent. Simple bisDRIP-seq and DRIP-seq correlation is not sufficient. BisDRIP-seq with in vitro RNase H treatment should be performed.

We fully agree with the reviewer's suggestion. We therefore performed an RNase H-control bisDRIP-seq experiment as suggested by the reviewer and as detailed below. We then demonstrated that R-loop signal was lost upon RNase H treatment. Additionally, we performed a matched control experiment to show that, absent RNase H, the same experiment with the same input material recapitulates our earlier results.

The RNase H treatment experiment was performed by repeating the initial bisDRIP-seq steps, including lysing the cells in the presence of bisulfite, degrading proteins with Proteinase K treatment and fragmenting the DNA. RNase I was added to deplete the sample of single-stranded RNAs that could bind the S9.6 antibody (Phillips et at., 2013). After this point, the sample was split into two. One sample received an RNase H treatment. This sample was treated with RNase H both immediately prior to RNA/DNA immunoprecipitation and during the final two wash steps of the immunoprecipitation. The other sample was treated in an identical manner, except that no RNase H was added to the sample. Sequencing libraries were then prepared from both samples using the bisDRIP-seq method.

After performing this experiment, we compared the RNase H and matched control libraries to the libraries generated by our other bisDRIP-seq experiments. Importantly, we show that the R-loops in the MALAT1, NEAT1, RPPH1 and the histone genes lack signal in the RNase H-treated sample (Figure 1—figure supplement 2, Figure 5—figure supplement 1, Figure 6—figure supplement 2).

To further demonstrate that RNase H depletes metaplot R-loop signal, the revised manuscript now includes metaplots of the bisDRIP-seq signal relative to transcription start sites from the RNase H and matched-control experiments (Figure 3—figure supplement 3). In our initial manuscript, we observed strand asymmetry in metaplots of bisDRIP-seq scores both 5' and 3' of the transcription start site. This asymmetry was interpreted in our initial manuscript as representing antisense R-loops and sense R-loops, respectively. In our new matched-control bisDRIP-seq experiment, we continue to observe this strand asymmetry (Figure 3—figure supplement 3). Importantly, strand asymmetry is lost in the sample treated with RNase H (Figure 3—figure supplement 3). This suggests that the observed strand asymmetry near transcription start sites is dependent on R-loops.

We thank the reviewer for this suggestion since this important control experiment strengthens our manuscript by validating that the observed bisDRIP-seq strand asymmetry is due to R-loops.

2) In bisDRIP-seq, signals not only come from R-loops, but also from promoter single-stranded structures. Therefore it is important to include non-denatured cytosine-uracil-conversions (without S9.6 pull-down) as bisDRIP-seq input.

We agree that sequencing the bisulfite-treated DNA used as input for the RNA-DNA hybrid immunoprecipitation helps ensure that artifacts are not contributing to our results.

To address this concern, we prepared sequencing libraries from two separate bisDRIP-seq “input” samples. These samples were treated identically to our bisDRIP-seq experiments, except that no bisDRIP-seq immunoprecipitation steps were performed on these samples. We then sequenced these two libraries using the bisDRIP-seq methodology.

We examined the amount of input bisDRIP-seq signal at individual R-loop forming regions. These regions included the RPPH1 gene, the histone gene loci, MALAT1 and NEAT1 promoter regions (Figure 1—figure supplement 2, Figure 5—figure supplement 1, Figure 6—figure supplement 2). In these R-loop forming region, almost no bisDRIP-seq signal was observed in the absence of S9.6 immunoprecipitation. This suggests that there is considerable enrichment of these R-loops by S9.6 immunoprecipitation.

As suggested by the reviewer, we also used the results of the input bisDRIP-seq experiments to perform background correction. In general, there was very little signal in promoter regions. As a result, the input background correction (Figure 1—figure supplement 2) did not significantly alter the shape of the metaplot. This result suggests that the high bisDRIP-seq scores observed near transcription start sites requires the S9.6 immunoprecipitation of RNA-DNA hybrids.

We agree with the reviewer that this important validation of our method should provide readers with additional confidence in our methodology.

3) Template-strand bisDRIP-seq scores were subtracted from non-template-strand bisDRIP-seq scores to artificially recover R-loop signals. However, no assay was used to prove the validity of such a method.

We apologize that our manuscript did not sufficiently discuss the basis for using bisDRIP-seq score asymmetry as evidence for the presence of R-loops. We agree that providing additional evidence that R-loops lead to preferential bisDRIP-seq strand asymmetry adds additional clarity to the manuscript.

We have taken two main steps to address the reviewer's suggestion.

First, as discussed above, we now specifically show that strand asymmetry is lost when the sample is treated with RNase H (Figure 3—figure supplement 3). This provides strong validation of our use of strand asymmetry as a measure of R-loop formation.

Second, we have improved the wording regarding the discussion of the theoretical and experimental basis for using strand asymmetry to map R-loops. Non-denaturing bisulfite mapping of R-loops is one of the best-established methods for mapping R-loops and is used to map R-loops at individual gene loci (Chédin, F., 2016). A critical aspect of this type of R-loop measurement is to show that bisulfite preferentially labels one strand relative to the other strand (Yu et al., 2003). We have included more of this discussion in the Results sections entitled “bisDRIP-seq concept” and “Transcription-dependent R-loops form downstream of transcription start sites” to give readers a better understanding of the historical basis for this approach.

We hope these changes will provide further assurance to readers that comparing the reactivity of each strand to bisulfite provides a method to predict the location of R-loops.

4) Since previously reported DRIPc-seq already provides strand-specific, near base-pair resolution (PMID: 27373332), basic comparisons should be performed to test whether bisDRIP-seq outperforms DRIPc-seq in a significantly manner.

In our initial manuscript, we largely focused on comparing our data to the more established DRIP-seq technique. However, we agree with the reviewer that there is considerable value to comparing our technique to DRIPc-seq. The revised manuscript now has a figure (Figure 7) specifically devoted to comparing our results with DRIPc-seq. These results are now discussed in the new section entitled “Comparison of bisDRIP-seq to existing R-loop mapping methods.” Additionally, we have made a number of smaller changes to the paper to highlight how our method advances R-loop mapping relative to existing methods.

In our new Results section, we discuss why DRIPc-seq, as performed on human cells in (PMID:27373332) by Sanz et al., is not expected to provide near-nucleotide resolution of R-loops. Most significantly, as described by Sanz et al., 2016, the technique is essentially a simple RNA immunoprecipitation sequencing (RIP-seq) experiment. Thus, both the RNA incorporated in the RNA-DNA hybrid and the surrounding RNA is immunoprecipitated by the S9.6 antibody and sequenced. This should reduce the resolution of DRIPc-seq to the full length of the RNA. The evidence that Sanz et al. provide for the method's improved resolution derives from the fact that they observe smaller peak sizes than they had observed with DRIP-seq. However, the strategy employed by Sanz et al., should theoretically map to the entire RNA and not just the portion of the RNA in an RNA-DNA hybrid. We believe this is important context prior to actually examining the data from each method.

In order to directly compare our data to the data from DRIPc-seq, we plotted a metaplot of DRIPc-seq signal overlapped with bisDRIP-seq R-loop signal (Figure 7). Our map of R-loop formation shows a tight peak of R-loop signal near the transcription start site. By contrast, DRIPc-seq sense-strand signal shows enrichment near transcription start sites but remains high 5 kb downstream of the transcription start site. This is consistent with a model where bisDRIP-seq maps the exact location of R-loops and DRIPc-seq maps the location of the RNAs that are partially incorporated into the R-loop.

We also compared the bisDRIP-seq R-loop map to the DRIPc-seq map in individual genes. In Figure 7 we observe that both methods map R-loops to similar locations within the MALAT1 gene. However, the R-loop boundary is clearly better demarcated with bisDRIP-seq. Similarly, Figure 7 shows that both methods map R-loops to the NEAT1 loci, but the boundary is better demarcated by bisDRIP-seq.

We also address whether the time at which the R-loop is mapped in the two protocols influences the resolution of R-loop mapping. In bisDRIP-seq, R-loop labeling commences at the cell lysis step since the lysis buffer contains bisulfite. In all other protocols, the location of R-loops is measured by immunoprecipitation several hours or days after cell lysis. It therefore remains possible that R-loops could expand, contract, or disappear ex vivo, which could affect the results of conventional R-loop mapping methods. While this issue is not discussed extensively in the modern literature, it was shown in the 1970s intron-mapping R-loop literature that R-loops can be unstable in solution (Kaback et al., 1979). Thus delaying the measurement of R-loops could allow R-loops to change their structure to be more thermodynamically favorable. Therefore, we performed a bisDRIP-seq protocol in which the bisulfite step is delayed until the second day of the protocol. In this protocol, we observed several examples where the borders of the R-loop were less well demarcated (Figure 7) and in some cases, R-loops could not be detected (see Figure 7). This experiment further highlights the value of bisDRIP-seq, which labels R-loops during cell lysis and is less likely to be affected by ex vivo changes in R-loops than conventional methods.

In addition to directly comparing our results with DRIPc-seq, we have added new figures that highlight how our method provides results that are unobtainable by DRIPc-seq. In Figure 6—figure supplement 3, we show examples of individual reads that display cytosines conversions on one side of the read, while no conversions are observed on the other side. Each of these reads likely derive from the border of an R-loops. Notably, these reads come from the MALAT1 R-loop-forming region. This type of resolution cannot be achieved using DRIPc-seq since a read can come from any part of the RNA associated with the R-loop: either the portion hybridized to the DNA or the portion of the RNA that is not hybridized to the DNA. Therefore, there is no way of knowing how the read is positioned relative to the R-loop.

We wish to thank the reviewer for suggesting that we compare our data with the data from DRIPc-seq. We believe that this comparison will help readers better understand how our method is different and, in many ways an improvement to existing methods.

Reviewer #2:The authors describe a new method to map the position of R-loops at higher resolution than existing techniques. The approach entails converting cytosines to uracils in the displaced DNA strand with bisulfite, while the cytosines in the opposite strand are protected by RNA. To minimize the amount of sequencing they focus on regions immunoprecipitated by the S9.6 antibody specific for RNA-DNA hybrids. After bioinformatic processing the authors obtain a genome-wide "bisDRIP-score" which measures the frequency of single-stranded DNA present at a given locus. Asymmetric single-stranded regions are considered R-loops. R-loops are correlated with transcription and enriched in the promoter region of active genes. The high resolution of bisDRIP-seq allowed the authors to identify the TSS as a 5' boundary and the first exon-intron junction as a 3' boundary. Finally, they report that histone genes and lncRNAs MALAT1 and NEAT1 have a particularly strong R-loop signature.R-loops are believed to play important roles in gene regulation and the manuscript by Dumelie et al. describes a creative strategy to better map their genomic localization. The results are well presented and the methods are described clearly and in much detail. Overall I support publication in eLife. There are some major issues that need to be addressed, mostly ensuring that the authors are truly measuring R-loops and that no biases are skewing their analyses.

The reviewer notes that “R-loops are believed to play important roles in gene regulation and the manuscript by Dumelie et al. describes a creative strategy to better map their genomic localization”. Additionally, the reviewer feels that “the results are well presented and the methods are described clearly and in much detail”. The reviewer therefore states that they “support publication in *eLife*”. The reviewer then suggests a few issues are important for the revision to address.

– BisDRIP score: I find the explanation of how this score was calculated rather convoluted. The choice of using the sum of the bisDRIP-score for cumulative anlayses seems questionable. This makes it impossible to compare axes when different number of regions are analyzed. For example 3C vs. 3E. Why not using the average?

First, we thank the reviewer for their positive comments regarding the overall clarity of our manuscript. However, we fully agree with the reviewer that we could have provided a less convoluted explanation for how we calculated bisDRIP-seq scores. We have updated our explanation of this technique in the Results section to make it more concise and clear (subsection “Near-Nucleotide Resolution Mapping of R-loops on a Genome-Wide Scale”). We have removed some of the details and moved them to the Materials and methods section, which now contains a more detailed explanation for how we calculated bisDRIP-seq scores. The Materials and methods section entitled “Calculation of raw bisDRIP-seq read scores” has also been edited in an effort to make it easier to understand. We hope these textual changes will assist readers in fully understanding our method.

The reviewer asks why we used sums rather than averages in the metaplot of genes. We feel that this is a valid concern and recognize that using the cumulative sum of genes' bisDRIP-seq scores makes some comparisons more challenging. To address the reviewer’s concern, we have now added an additional figure supplement (Figure 3—figure supplement 1) where mean bisDRIP-seq scores are plotted rather than the cumulative sum. As can be seen, the overall result is very similar when using averages rather than sums. Nevertheless, we hope that this additional figure supplement will allow readers to more directly compare the mean bisDRIP-seq scores in this metaplots of Figure 3.

Our main rationale for presenting the cumulative sum of bisDRIP-seq scores in the panels of Figure 3 was to show that there is no change in the absolute differences in bisDRIP-seq scores between Figure 3. This demonstrates to readers that the bisDRIP-seq asymmetry observed in the metaplot of all promoters (Figure 3) is due exclusively to the active promoters. Outside of that, we believed that the most important aspect of our plots were their shape of the metaplot lines. The shape of the metaplot lines are not affected by whether we use cumulative scores or mean scores, since the transformation is applied evenly to all points (i.e.: the mean at each point is simply the cumulative sum divided by the number of genes).

– Validations: because bisDRIP-seq is a new genome-wide technique a high level of confidence in its measurements is required before it can be employed for new discoveries. 1) The authors show significant correlation genome-wide between DRIP-seq (the established technique) and their new bisDRIP-seq, but the correlation is obviously not perfect. Can the authors identify examples in which their new technique and the old technique differ and independently verify that the new mapping is more accurate? 2) To validate DRIP-seq typically RNaseH is used to remove RNA-DNA hybrids and prove the specificity of the signal. I believe this would be a more convincing control than triptolide also for bisDRIP-seq.

We appreciate these suggestions, which were also raised by reviewer #1. Please see our response to reviewer #1, point 4, for our detailed comparison of bisDRIP-Seq to DRIPc-seq. Also, please see our response to reviewer #1, point 1 for our discussion of how our new experiments using RNase H support the specificity of our method.

– Boundaries: the authors state that the TSS and the first exon-intron junction form 5' and 3' boundaries for R-loop formation, respectively, yet in several figures (e.g. Figure 3, Figure 4) I see signal above background upstream and downstream of these boundaries. Is this because bisDRIP-seq is not exactly single-base resolution, because of smoothing artifacts, or because these are not hard boundaries? Also, it was reported that restriction enzyme digestion can lead to overrepresentation of ORFs and first exon bias (László Halász et al., Genome Res 2017). It is important that the authors exclude this possibility in their own analyses.

We really appreciate the reviewer's comment. In the original manuscript, we only displayed bisDRIP-seq scores in metaplots. As a result, we did not fully use the benefits of our technique to display the location of R-loop boundaries. We have added new figures that more clearly show the location of the exon-intron boundary, which we believe will address the reviewer's comment regarding R-loop boundaries.

The reviewer is correct to note that there is R-loop signal 5' of the transcription start site. In regard to this comment, it is important to note that the location of transcription start sites are not well defined. This is because genes do not necessarily have a single site at which they begin transcription. Instead, they often begin transcription at numerous sites within a promoter region (Carninci et al., 2006). We discuss this briefly in subsection “Transcription-dependent R-loops form downstream of transcription start sites”. This uncertainty in the location of the transcription start sites likely contributes to the R-loop signal observed upstream of the transcription start site.

The reviewer is also correct to note that there is R-loop signal 3' of the first exon-intron junction. In contrast to transcription start sites, exon-intron junctions are well-defined. As such, our method should be able to observe a sharp R-loop boundary in relation to exon-intron junctions. A likely explanation for the observed R-loop signal 3' of the exon-intron junction is our use of bisDRIP-seq scores in our metaplots. Importantly, the bisDRIP-seq scores have read length resolution. In other words, the entire read is assigned a bisDRIP-seq score, even if the read crosses the border of the R-loop. This will reduce the resolution in the metaplots. This was not described clearly enough in the original manuscript, for which we apologize to the reviewer.

In order to allow readers to see the R-loop boundary at near-nucleotide resolution, we have added a metaplot of cytosine-conversion data. In this metaplot, we examine cytosine-conversion strand-asymmetry relative to the first exon-intron junction. Cytosine-conversion strand-asymmetry is defined as the ratio of the number of conversions on the non-template strand relative to the template strand after background correction. Unlike the bisDRIP-seq score metaplot, this plot should provide near-nucleotide resolution. Indeed, we observe with this plot that the strand asymmetry in cytosine conversions is lost within base pairs of the first exon-intron junction (Figure 4).

We hope that this result provides additional clarity to readers regarding the location of the 3' R-loop boundary. Additionally, we believe this plot nicely highlights the capability of bisDRIP-seq to map R-loops at near-nucleotide resolution.

The reviewer also commented on the first exon bias observed by Halász et al., 2017, in DRIP-seq experiments. Halász et al. found that the sequence composition of first exons is biased so that this region is less likely to be fragmented by the restriction enzymes used in DRIP-seq approaches. As a result, DRIP-seq is biased towards immunoprecipitating first exons. We have added commentary to the Discussion section (first paragraph) regarding this issue. In particular, this first exon bias should result in higher bisDRIP-seq signal on both the template and nontemplate strands of DNA in the first exon. However, this should not lead to strand asymmetry between the signal on either strands of DNA. Since most of our conclusions are derived from strand asymmetry, we do not believe that the conclusions made in this manuscript should be significantly affected by the bias uncovered by Halász et al., 2017. We agree with the reviewer that this discussion is important to include in the manuscript and appreciate the thoughtful advice that they have provided here.

– Methylcytosines: it seems that the presence of methylated cytosines would make it difficult to detect R-loops in heavily methylated areas of the genome. Is this a problem? It should be discussed.

We agree that this is an important issue to discuss regarding our method. We have added discussion regarding this issue to the bisDRIP-seq protocol portion of the Methods section.

– Genes with strong R-loop signals: the authors conclude that not having introns is a reason why some of the genes make in the top 25 list shown in Table 1 but alternative possibilities should be considered: 1) MALAT1, NEAT1 and histone RNAs are not terminated canonically; has that anything to do with R-loops? 2) What is the relationship between steady state RNA levels and formation of R-loops? GRO-seq measures transcription levels, and a weak correlation is visible in Figure 5, but what about total RNA levels? Ribominus RNA-seq to measure steady-state levels of nuclear RNAs might be more suitable for this.

We agree with the reviewer that the non-canonical transcription termination observed in histone genes, *MALAT1*, *NEAT1* and *RPPH1* may contribute to R-loop formation in these genes. Indeed, we think this a very interesting possibility. We have added additional commentary to the Discussion portion of the manuscript to discuss this possibility.

We also agree with the reviewer that steady state levels of RNAs are a potential explanation for genes having high R-loop signal. As suggested by the reviewer we compared R-loop signal to the RNA-seq levels of nuclear RNA from an MCF-7 RNA-seq dataset generated by ENCODE (Figure 5—figure supplement 1). Perhaps not surprisingly, RNA levels correlate slightly worse with R-loop levels than R-loops had correlated with GRO-seq data. This seems reasonable since R-loops appear to form with newly transcribed RNA. Nonetheless, we believe that adding this comparison makes our manuscript more complete and will address the concerns of many readers.

Reviewer #3:In this manuscript, Dumelie et al. develop a new method, Bis-DRIP Seq, to identify the genomic localization of R loops at near-nucleotide resolution. Given the growing interest in understanding the function of R loops, this new method will benefit research in this area. Using this technique authors show that R loops are associated with the promoter regions of genes with or without introns. However, in genes with introns, 1st exon-intron junction defines the boundary of the R loop. We suggest a few minor experiments before publication of this manuscript.

The reviewer observes that there is “growing interest in understanding the function of R-loops” and they state that bisDRIP-seq “will benefit research in this area.” The reviewer therefore suggests “a few minor experiments before publication of this manuscript.”

Specific comments:1) The authors highlight that BisDRIP-Seq provides more information about R loop boundaries as compared to previous DRIP-Seq datasets. This new method is an extension of the previously established DRIP-Seq and in theory it is understandable why it is an improvement. But since this is a 'methodology paper', the authors must provide comparative evidence that it is better. A figure showing how their dataset compares with a DRIP-Seq dataset will be beneficial.

We thank the reviewer for this question. This was raised by the other reviewers and a detailed description of our new comparisons is provided in our response reviewer #1, point 4.

2) The authors show that the 1st exon-intron junction serves as the 3' R-loop boundary (Figure 4) which they suggest might be due RNA splicing (Figure 7). To extend the biology a little more, the authors should look at a few genes that are known to have alternative splice forms to test if R loop boundaries show differences in genes that are known to have alternative splicing versus those that do not.

We agree with the reviewer that this would be a very interesting possibility. We were not able to identify genes that have alternative splice forms with clear R-loop signal past the first exon. This may be a result of us not having good annotation of which genes have retained first introns in MCF-7 cells. Another possibility is that the cell may target mRNAs with retained first-introns that form R-loops to be degraded by RNase H. In this model, mRNAs with retained introns that are observable by RNA-seq must have mechanisms to prevent the RNA from forming a long R-loop that would be susceptible to RNase H degradation. We hope that we can better answer the reviewer's question in our future studies of R-loops.

We would like to thank all of the reviewers for taking the time and effort to provide ideas for enhancing our manuscript. We believe that these suggestions and experiments have helped make the manuscript more compelling to *eLife* readers, while also strengthening our original conclusions.

Reference

Schwalb, B., Michel, M., Zacher, B., Frühauf, K., Demel, C., Tresch, A., Gagneur, J. and Cramer, P., 2016. TT-seq maps the human transient transcriptome. Science, 352, 1225-1228.